# Jaccard Metric Losses: Optimizing the Jaccard Index with Soft Labels

**Zifu Wang**[1][*]          **Xuefei Ning**[2]          **Matthew B. Blaschko**[1]

[1] ESAT-PSI, KU Leuven, Leuven, Belgium
[2] Department of Electronic Engineering, Tsinghua University, Beijing, China

## Abstract

Intersection over Union (IoU) losses are surrogates that directly optimize the Jaccard index. Leveraging IoU losses as part of the loss function have demonstrated superior performance in semantic segmentation tasks compared to optimizing pixel-wise losses such as the cross-entropy loss alone. However, we identify a lack of flexibility in these losses to support vital training techniques like label smoothing, knowledge distillation, and semi-supervised learning, mainly due to their inability to process soft labels. To address this, we introduce Jaccard Metric Losses (JMLs), which are identical to the soft Jaccard loss in standard settings with hard labels but are fully compatible with soft labels. We apply JMLs to three prominent use cases of soft labels: label smoothing, knowledge distillation and semi-supervised learning, and demonstrate their potential to enhance model accuracy and calibration. Our experiments show consistent improvements over the cross-entropy loss across 4 semantic segmentation datasets (Cityscapes, PASCAL VOC, ADE20K, DeepGlobe Land) and 13 architectures, including classic CNNs and recent vision transformers. Remarkably, our straightforward approach significantly outperforms state-of-the-art knowledge distillation and semi-supervised learning methods. The code is available at https://github.com/zifuwanggg/JDTLosses.

## 1 Introduction

The Jaccard index, also known as the intersection over union (IoU), is a widely used metric in the evaluation of semantic segmentation models. Its appeal lies in its scale invariance and its superior ability to reflect the perceptual quality of a model compared to pixel-wise accuracy [17, 40]. In line with the principles of empirical risk minimization in statistical learning theory, the metric used for evaluation should also be optimized during training [60]. Consequently, directly optimizing IoU via differentiable surrogates has found considerable attention in the literature [46, 49, 2, 72, 17, 73].

Notably, IoU is only defined when both predictions and ground-truth labels are discrete binary values, i.e., they reside on the vertices of a $p$-dimensional hypercube $\{0, 1\}^p$. However, the output of neural networks are soft probabilities that are in the interior of the hypercube $[0, 1]^p$. To extend the value of IoU from vertices to the entire hypercube, two popular strategies are currently in use. The first strategy relaxes set counting with norm functions, exemplified by the soft Jaccard loss (SJL) [46, 49] and the soft Dice loss (SDL) [57, 17]. The second strategy computes the Lovasz extension of the IoU, such as the Lovasz hinge loss [72] and the Lovasz-Softmax loss (LSL) [2]. These losses facilitate a plug-and-play use and have significantly enhanced the performance of segmentation models, outperforming pixel-wise losses such as the cross-entropy loss (CE) and the focal loss [35]. For instance, Rakhlin et al. [50] won the land cover segmentation task of the DeepGlobe Challenge [12] utilizing LSL. Additionally, SDL is becoming increasingly prominent in recent segmentation works [5, 9, 8, 29], such as Segment Anything [29]. Moreover, SJL and SDL are now the standard for training medical imaging models [17, 27].

---

[*]Correspondence to: zifu.wang@kuleuven.be

Despite their widespread application, current IoU losses lack the flexibility needed to accommodate key training techniques. Specifically, while these losses relax predictions from $\{0, 1\}^p$ to $[0, 1]^p$, they neglect the possibility of labels residing in $[0, 1]^p$, making them incompatible with soft labels. Soft labels have been employed in numerous state-of-the-art models and have proven effective in enhancing network accuracy [37, 68, 62] and calibration [20, 44, 42, 82]. For example, label smoothing (LS) [58] generates soft labels by taking a weighted average of one-hot hard labels and a uniform distribution over labels. In knowledge distillation (KD) [22], class probabilities generated by a teacher network serve as soft labels to guide a student model. In semi-supervised learning (SSL) such as MixMatch [4], $k$ views of an unlabeled image are fed to a classifier, and the predictions are averaged to create soft labels.

Motivated by the limitation of existing IoU losses, particularly their inability to accommodate soft labels and, consequently, techniques like LS, KD, and SSL, we propose two variants of SJL, termed Jaccard metric losses (JMLs) since they are metrics on $[0, 1]^p$. JMLs yield the same value as SJL on hard labels but are fully compatible with soft labels. Therefore, we can safely replace the existing implementation of SJL with JMLs without affecting the performance on hard labels. To embed our losses with use cases (LS, KD, SSL), we first introduce boundary label smoothing (BLS) which facilitates the integration of label smoothing into the segmentation task. BLS can be used independently and also synergizes with KD and SSL. Subsequently, we present a confidence-based scheme for selecting classes to contribute to the loss computation, thereby enhancing performance in KD. We conduct extensive experiments on 4 datasets (Cityscapes [11], PASCAL VOC [18], ADE20K [81], DeepGlobe Land [12]) across 13 architectures, including classic CNNs [21, 54] and recent vision transformers [69]. Our results demonstrate significant improvements over the cross-entropy loss. Moreover, our straightforward approach outperforms state-of-the-art segmentation KD and SSL methods by a substantial margin.

## 2 Methods

### 2.1 Preliminaries

Given a segmentation output $\dot{x} \in \{1, ..., C\}^p$ and a ground-truth $\dot{y} \in \{1, ..., C\}^p$ where $C$ is the number of classes, for each class $c$, we define the set of predictions as $x^c = \{\dot{x} = c\}$, the set of ground-truth as $y^c = \{\dot{y} = c\}$, the union as $u^c = x^c \cup y^c$, the intersection as $v^c = x^c \cap y^c$, the symmetric difference (the set of mispredictions) as $m^c = (x^c \setminus y^c) \cup (y^c \setminus x^c)$, and the Jaccard index as $\text{IoU}^c = |v^c|/|u^c|$. For multi-class segmentation, $\text{IoU}^c$ are averaged across classes, yielding the mean IoU (mIoU). In the sequel, we will encode sets as binary vectors $x^c, y^c, u^c, v^c, m^c \in \{0, 1\}^p$ where $p$ is the number of pixels, and denote $|x^c| = \sum_{i=1}^p x_i^c$ the cardinality of the corresponding set. For simplicity, we will drop the superscript $c$ in the following.

In order to optimize IoU in a continuous setting, we need (almost everywhere) differentiable interpolations of this discrete score. In particular, we want to extend the IoU loss

$$\Delta_{\text{IoU}} : x \in \{0, 1\}^p, y \in \{0, 1\}^p \mapsto 1 - \frac{|v|}{|u|} = \frac{|m|}{|y \cup m|} \tag{1}$$

with $\overline{\Delta}_{\text{IoU}}$ so that it attains a value with any vector of predictions $\tilde{x} \in [0, 1]^p$. In what follows, when the context is clear, we will use $x$ and $\tilde{x}$ interchangeably.

The soft Jaccard loss (SJL) [46, 49] generalizes IoU by realizing that when $x, y \in \{0, 1\}^p$, $|v| = \langle x, y \rangle$ and $|u| = |x| + |y| - |v| = \|x\|_1 + \|y\|_1 - \langle x, y \rangle$. Therefore, SJL replaces the set notation with vector functions:

$$\overline{\Delta}_{\text{SJL},L^1} : x \in [0, 1]^p, y \in \{0, 1\}^p \mapsto 1 - \frac{\langle x, y \rangle}{\|x\|_1 + \|y\|_1 - \langle x, y \rangle}. \tag{2}$$

The $L^1$ norm can be replaced with the squared $L^2$ norm [17]:

$$\overline{\Delta}_{\text{SJL},L^2} : x \in [0, 1]^p, y \in \{0, 1\}^p \mapsto 1 - \frac{\langle x, y \rangle}{\|x\|_2^2 + \|y\|_2^2 - \langle x, y \rangle}. \tag{3}$$

## 2.2 The Limitation of Existing IoU Losses

The primary shortcoming of current IoU losses is that they do not necessarily have desired properties when presented with soft labels, i.e., when $y \in [0, 1]^p$. This limitation impedes their application in crucial training techniques like LS, KD, and SSL.

Consider the case of $\overline{\Delta}_{\mathrm{SJL},L^1}$, and for simplicity, a single pixel scenario: $\overline{\Delta}_{\mathrm{SJL},L^1} = 1 - \frac{xy}{x+y-xy}$. It is easy to confirm that for any $y > 0$, $\overline{\Delta}_{\mathrm{SJL},L^1}$ is minimized at $x = 1$ since it monotonically decreases as a function of $x$. Hence, $\overline{\Delta}_{\mathrm{SJL},L^1}$ is in general not minimized when $x = y$, a basic property anticipated from a loss function. Further analysis for high-dimensional cases and additional experiments on real datasets are provided in Appendix C and E, respectively.

$\overline{\Delta}_{\mathrm{SJL},L^2}$ does not exhibit this issue, since $\overline{\Delta}_{\mathrm{SJL},L^2} = 0 \Leftrightarrow |x|_2^2 + |y|_2^2 - 2\langle x, y\rangle = 0 \Leftrightarrow x = y$. However, it is known to yield inferior results compared to its $L^1$ counterpart, possibly due to its flatter nature around the minimum [17]. In practice, it is rarely used. For instance, in SMP [25], a popular open-source semantic segmentation project, only the $L^1$ version is implemented. Our evaluations in Appendix E confirm its inferior performance relative to the $L^1$ version. Additionally, the approach of substituting set notation with the $L^1$ norm is widely utilized in numerous other works, including the soft Dice loss [57, 17], the soft Tversky loss [53], the focal Tversky loss [1], and others. The soft Dice loss is also included in the formulation of the PQ loss [61] which is used in panoptic segmentation [28]. Consequently, all of them struggle with soft labels.

Losses based on the Lovasz extension, such as the Lovasz-Softmax loss [2], the Lovasz hinge loss [72], and the PixIoU loss [73], cannot handle soft labels as the Lovasz extension is not well-defined for $y \in (0, 1)^p$. More details and comparisons with the Lovasz-Softmax loss can be found in Appendix D and E, respectively. Automatically searched loss functions, such as Auto Seg-Loss [33] and AutoLoss-Zero [32], also fail to accommodate soft labels as their search space is confined to integral labels.

In summary, despite their widespread adoption in recent works on semantic segmentation [17, 27, 9, 8, 29] and panoptic segmentation [5, 61, 74, 75], these losses all exhibit a common shortcoming: an inability to handle soft labels. In this paper, we specifically concentrate on re-designing SJL. Other losses, including the soft Dice loss, the soft Tversky loss, and the focal Tversky loss, are addressed in our subsequent work [66].

## 2.3 Jaccard Metric Losses

We can rewrite the intersection $|v|$ and the union $|u|$ as a function of the set difference $|m|$:

$$|v| = \frac{1}{2}(|x| + |y| - |m|) \text{ and } |u| = |v| + |m|. \tag{4}$$

Note that $|m| = \|x - y\|_1$. Combining these yields:

$$|v| = \langle x, y\rangle = \frac{1}{2}(\|x + y\|_1 - \|x - y\|_1), \tag{5}$$

$$|u| = \langle x, y\rangle + \|x - y\|_1 = \frac{1}{2}(\|x + y\|_1 + \|x - y\|_1), \tag{6}$$

where the equalities hold when $x, y \in \{0, 1\}^p$.

After eliminating erroneous combinations that have the same issue as SJL, we are left with two candidates $\overline{\Delta}_{\mathrm{JML},1}, \overline{\Delta}_{\mathrm{JML},2} : [0, 1]^p \times [0, 1]^p \to [0, 1]$ that are defined as:

$$\overline{\Delta}_{\mathrm{JML},1} = 1 - \frac{\|x + y\|_1 - \|x - y\|_1}{\|x + y\|_1 + \|x - y\|_1}, \tag{7}$$

$$\overline{\Delta}_{\mathrm{JML},2} = 1 - \frac{\langle x, y\rangle}{\langle x, y\rangle + \|x - y\|_1}. \tag{8}$$

It is a well-known result that $\Delta_{\mathrm{IoU}}$ is a metric on $\{0, 1\}^p$ [30]. In Theorem 2.1 (see Appendix F for the proof), we show that both $\overline{\Delta}_{\mathrm{JML},1}$ and $\overline{\Delta}_{\mathrm{JML},2}$ are also metrics on $[0, 1]^p$. Therefore, we call them Jaccard Metric Losses (JMLs).

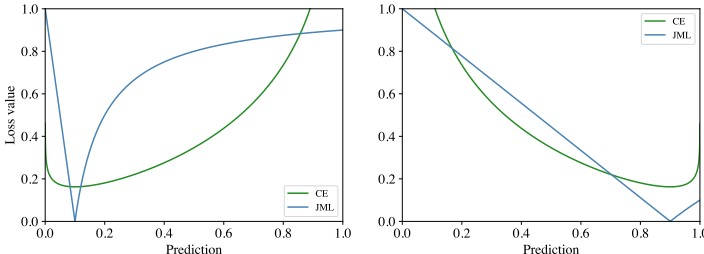

Figure 1: Loss value vs. prediction with $y = 0.1$ (left) and $y = 0.9$ (right).

**Theorem 2.1.** *Both $\overline{\Delta}_{JML,1}$ and $\overline{\Delta}_{JML,2}$ are metrics on $[0,1]^p$. Neither $\overline{\Delta}_{SJL,L^1}$ nor $\overline{\Delta}_{SJL,L^2}$ is a metric on $[0,1]^p$.*

Recall the definition of a metric:

**Definition 2.2** (Metric [14]). A mapping $f : M \times M \to \mathbb{R}$ is called a metric on $M$ if for all $a, b, c \in M$, it satisfies the following conditions: (i) (reflexivity). $f(a,a) = 0$. (ii) (positivity). $a \neq b \implies f(a,b) > 0$. (iii) (symmetry). $f(a,b) = f(b,a)$. (iv) (triangle inequality). $f(a,c) \leq f(a,b) + f(b,c)$.

Having a loss function $\overline{\Delta}$ that is a metric carries numerous benefits. Reflexivity and positivity collectively imply that $\forall x, y \in [0,1]^p, x = y \Leftrightarrow \overline{\Delta} = 0$, meaning that $\overline{\Delta}$ would be compatible with soft labels. The triangle inequality also provides insightful guidance. Applied to KD, it yields:

$$\overline{\Delta}(S, L) \leq \overline{\Delta}(S, T) + \overline{\Delta}(T, L). \tag{9}$$

Here, $S$ represents the student model, $T$ the teacher model, and $L$ the ground-truth labels. In the KD, we initially train the teacher model using the ground-truth labels, and then minimize the loss between the teacher and the student model. Equation (9) suggests that if we adhere to this process—equivalent to minimizing the right-hand side of the equation—we also minimize the upper bound on the student model's loss with respect to the ground-truth labels.

Furthermore, as Theorem 2.3 shows (see Appendix G for the proof), when only hard labels are involved, $\overline{\Delta}_{JML,1}, \overline{\Delta}_{JML,2}$ and $\overline{\Delta}_{SJL,L^1}$ are identical. Hence, we can safely replace the existing implementation of SJL with JMLs. When soft labels are introduced, $\overline{\Delta}_{JML,1}$ and $\overline{\Delta}_{JML,2}$ might yield different values. We discuss their distinctions in Appendix H. From both theoretical and empirical perspectives, we find that $\overline{\Delta}_{JML,1}$ is slightly more favorable than $\overline{\Delta}_{JML,2}$. As a default in our experiments, we use $\overline{\Delta}_{JML,1}$.

**Theorem 2.3.** $\forall x \in [0,1]^p, \ y \in \{0,1\}^p$ *and* $x \in \{0,1\}^p, \ y \in [0,1]^p, \ \overline{\Delta}_{JML,1} = \overline{\Delta}_{JML,2} = \overline{\Delta}_{SJL,L^1}.$ $\exists x, y \in [0,1]^p, \overline{\Delta}_{JML,1} \neq \overline{\Delta}_{JML,2} \neq \overline{\Delta}_{SJL,L^1}.$

Figure 1 plots the loss value of $\overline{\Delta}_{JML,1}$ and the cross-entropy loss (CE) for soft labels $y = 0.1$ (left) and $y = 0.9$ (right). Typically, the model is randomly initialized, resulting in a low initial confidence. Although the gradient of CE is substantial at the beginning, it plateaus as training progresses and the prediction is close to the target. Conversely, $\overline{\Delta}_{JML,1}$ offers effective supervision later in the process. However, when $y$ is extremely small (e.g., $y = 0.1$), $\overline{\Delta}_{JML,1}$ can become overly steep, potentially causing problems in KD. This issue will be explored in greater detail in Section 2.4.2.

## 2.4 Use Cases

Soft labels find wide-ranging applications. In this paper, we investigate three of the most common use cases: label smoothing (LS), knowledge distillation (KD) and semi-supervised learning (SSL). We leave the discussion on SSL in Appendix I.

### 2.4.1 Label Smoothing

In LS [58], the one-hot encoding is combined with a uniform distribution under a smoothing coefficient $\epsilon$, resulting in soft labels $SL^\epsilon$. In JML-LS, a network $H$ is trained with the following loss:

$$\mathcal{L}_{\text{JML-LS}} = \lambda_{\text{CE}}\mathcal{L}_{\text{CE, LS}} + \lambda_{\text{JML}}\mathcal{L}_{\text{JML, LS}} \tag{10}$$

such that $\mathcal{L}_{\text{CE, LS}} = \text{CE}(H, SL^{\epsilon})$ and $\mathcal{L}_{\text{JML, LS}} = \text{JML}(H, SL^{\epsilon})$.

LS provides a regularization effect during training. However, when we regard semantic segmentation as a pixel-wise classification task and apply LS to every pixel in the image without distinction, we fail to consider spatial differences. This is particularly evident with SegFormer-B3 on ADE20K, which achieves an overall pixel-wise accuracy of $81.91 \pm 0.02\%$, but drops to $47.98 \pm 0.01\%$ in the boundary region. These areas, due to their inherent ambiguity, require stronger regularization. Moreover, our empirical findings suggest that smoothing non-boundary regions only yields marginal improvements. Therefore, we introduce boundary label smoothing (BLS), which only applies smoothing to labels near the boundary. We refer to the resulting loss as JML-BLS. Specifically, for every pixel $i$, we examine its $k \times k$ neighborhood $N_i$, and the pixel $i$ is regarded as a boundary pixel if there exists a pixel $j \in N_i$ such that their ground-truth labels are different: $y_i \neq y_j$. This can be efficiently computed by applying a max pooling layer to the one-hot encoding.

### 2.4.2 Knowledge Distillation

In KD [22], besides the ground-truth label $L$, a student model $S$ is trained to minimize the discrepancy to a teacher network $T$ simultaneously:

$$\mathcal{L}_{\text{CE, KD}} = \mu_{\text{L}}\mathcal{L}_{\text{CE, L}} + \mu_{\text{T}}\mathcal{L}_{\text{CE, T}} \tag{11}$$

where $\mathcal{L}_{\text{CE, L}} = \text{CE}(S, L)$ and $\mathcal{L}_{\text{CE, T}} = \text{CE}(S, T)$.

In JML-KD, we add additional supervision from JML:

$$\mathcal{L}_{\text{JML, KD}} = \nu_{\text{L}}\mathcal{L}_{\text{JML, L}} + \nu_{\text{T}}\mathcal{L}_{\text{JML, T}} \tag{12}$$

where $\mathcal{L}_{\text{JML, L}} = \text{JML}(S, L)$ and $\mathcal{L}_{\text{JML, T}} = \text{JML}(S, T)$.

Combining these, we have

$$\mathcal{L}_{\text{JML-KD}} = \lambda_{\text{CE}}\mathcal{L}_{\text{CE, KD}} + \lambda_{\text{JML}}\mathcal{L}_{\text{JML, KD}}. \tag{13}$$

Existing segmentation KD methods rely on pixel-wise supervision either in the output [36, 70] or in the feature space [64, 55]. Our $\mathcal{L}_{\text{JML, T}}$, however, enables direct distillation of the teacher's IoU information to the student which aligns with the final evaluation metric.

Moreover, recall that JMLs are averaged for each class, making it crucial to determine which classes contribute to the loss value at each iteration. We refer to these classes as *active classes*. In KD, the student uses information from non-target classes to learn the class relationship from the teacher [59, 77]. However, a teacher network may also include noisy and unhelpful predictions for unimportant classes. Therefore, we propose to filter out unimportant classes based on the confidence level of the teacher network. Specifically, JMLs are computed only over classes where the teacher's confidence exceeds a predefined threshold. This approach has two benefits: first, the student will not be distracted by irrelevant classes, aiding in generalization. Second, as shown in Figure 1, we find that the loss value of JMLs can become excessively steep around the target when the ground-truth is low (e.g. $y = 0.1$). Therefore, ignoring these classes can stabilize the training process.

We have also observed that the student benefits from a teacher trained with JML-BLS. In contrast, in classification tasks, it has been found that a teacher trained with LS can detrimentally affect the student's performance [44]. This is often cited as a counterexample to the notion that a more accurate teacher will necessarily distill a better student. We believe that the teacher, trained through JML-BLS, acquires intricate boundary information, and this "dark knowledge" is implicitly passed to the student, resulting in a more accurate student.

## 3 Experiments

**Experimental setups.** We adopt Pytorch Image Models (timm) [67], which provides implementations and ImageNet [13] pre-trained weights for various backbones. In our experiments, CNN backbones include ResNet-101/50/18 [21] and MobileNetV2 [54]; transformer backbones contain MiT-B5/B3/B1/B0 [69]. Segmentation methods consist of DeepLabV3+ [7], DeepLabV3 [6], PSPNet

[78], UNet [51] and SegFormer [69]. We evaluate models on Cityscapes [11], PASCAL VOC [18], ADE20K [81] and DeepGlobe Land [12]. In summary, 13 models and 4 datasets are studied in the sequel. More details of each model are in Appendix A and training recipes are in Appendix B.

**Evaluation metrics.** To provide a comprehensive comparison, we report both overall pixel-wise accuracy (Acc) and mean intersection over union (mIoU). To evaluate the effects of our methods on calibration, we present the expected calibration error (ECE) [20] and the ECE computed only over the boundary region, denoted as BECE. For Cityscapes, PASCAL VOC, and ADE20K, we repeat the experiments 3 times (except for SSL experiments that are single runs) and report performance on the validation set. For DeepGlobe Land, we conduct 5-fold cross-validation. All results are presented in the format of mean±standard deviation. We do not apply any test-time augmentation.

## 3.1 Results on Accuracy

We report results on Cityscapes (Table 1), PASCAL VOC (Table 2), ADE20K (Table 3), and DeepGlobe Land (Table 13, Appendix J). Key takeaways from our results are as follows:

**JML significantly improves accuracy (Acc and mIoU).** Compared to training with CE alone, incorporating JML as part of the loss function can significantly enhance a model's mIoU. The improvement is typically more than 2% on Cityscapes, PASCAL VOC, and ADE20K. Additionally, it can also increase a model's Acc. This suggests that the benefits of JML are not solely due to its alignment with the evaluation metric (mIoU), but also because it aids the overall optimization process.

**JML benefits more from soft labels.** For instance, the improvements of mIoU for DL3-R101 on PASCAL VOC are 0.48% (CE vs. CE-BLS) and 1.25% (JML vs. JML-BLS), respectively. We adopted training procedures that are heavily optimized for CE, which might explain why JML requires stronger regularization. More qualitative results can be found in Figure 7 and Figure 8 (Appendix L).

**KD is more effective than BLS, while BLS performs well on datasets with simple boundary condition.** Both BLS and KD consistently improve the model's accuracy. Despite its simplicity, BLS can yield significant improvements, especially on PASCAL VOC where boundary condition is less complex. As boundary condition becomes more intricate, as in Cityscapes and ADE20K, KD, seen as learned label smoothing [76], typically outperforms BLS.

**SOTA results on segmentation KD.** Without bells and whistles, our simple approach that only uses soft labels greatly exceeds SOTA segmentation KD methods, as shown in Table 4. Indeed, the model trained only with hard labels in JML already achieves a comparable result as some of these methods. For example, on PASCAL VOC, DL3-R18 achieves $74.42 \pm 0.52\%$ mIoU while CIRKD [70], a complex distillation method only attains $74.50\%$ mIoU. Note that our baseline is not stronger than theirs: our DL3-R18 trained with CE has $72.47 \pm 0.33\%$ mIoU while their number is $73.21\%$ (because we use a smaller batch size, see Appendix B).

**SOTA results on segmentation SSL.** By incorporating JML-BLS with AugSeg [80] (refer to Appendix I for further details), we achieve SOTA segmentation SSL results as illustrated in Table 11 (Appendix I). Notably, improvements are particularly significant on smaller splits - exceeding a 4% enhancement over AugSeg on the 92 split.

## 3.2 Results on Calibration

**CE-BLS and CE-KD sometimes hurt calibration.** Contrary to the common belief in classification [44, 42, 82], we find that both CE-BLS and CE-KD can sometimes deteriorate model calibration. For instance, in Cityscapes experiments, the lowest ECE is usually obtained with CE only. Nevertheless, CE-BLS can still improve model calibration near the boundary.

**JML compromises calibration.** Although the soft Dice loss can significantly improve a model's segmentation performance, it is well known to yield poorly calibrated models [41, 3]. We confirm that this is also the case for JML. Interestingly, while models trained with JML exhibit inferior top-class calibration as measured by ECE, they actually achieve better multi-class calibration as indicated by the static calibration error (SCE). We provide more detailed results on calibration in Appendix K.

**JML-BLS and JML-KD consistently improve calibration.** Although JML presents challenges in model calibration, these can be greatly mitigated by training models with soft labels. In JML experiments, both JML-BLS and JML-KD reliably enhance model calibration.

Table 1: Results on Cityscapes (%). Best results within CE and JML groups are highlighted in red and green, respectively. Best results across CE and JML groups are underscored.

| Model | Metric | CE | CE-BLS | CE-KD | JML | JML-BLS | JML-KD |
|---|---|---|---|---|---|---|---|
| DL3-R101 | Acc ↑ | 96.10 ± 0.02 | 96.11 ± 0.03 | - | 96.10 ± 0.04 | 96.25 ± 0.03 | - |
| | mIoU ↑ | 78.67 ± 0.32 | 78.70 ± 0.26 | - | 80.29 ± 0.17 | 80.66 ± 0.24 | - |
| | ECE ↓ | 0.76 ± 0.05 | 0.97 ± 0.04 | - | 2.74 ± 0.09 | 2.09 ± 0.02 | - |
| | BECE ↓ | 16.14 ± 0.22 | 11.59 ± 0.12 | - | 30.78 ± 0.19 | 21.20 ± 0.07 | - |
| DL3-R50 | Acc ↑ | 95.77 ± 0.08 | 95.83 ± 0.08 | - | 95.93 ± 0.04 | 96.05 ± 0.02 | - |
| | mIoU ↑ | 76.45 ± 0.68 | 77.02 ± 0.82 | - | 78.68 ± 0.42 | 79.10 ± 0.35 | - |
| | ECE ↓ | 0.62 ± 0.02 | 1.00 ± 0.03 | - | 2.84 ± 0.02 | 2.18 ± 0.01 | - |
| | BECE ↓ | 16.23 ± 0.16 | 11.67 ± 0.18 | - | 31.12 ± 0.10 | 20.51 ± 0.12 | - |
| DL3-R18 | Acc ↑ | 95.28 ± 0.02 | 95.31 ± 0.05 | 95.40 ± 0.04 | 95.46 ± 0.04 | 95.49 ± 0.05 | 95.59 ± 0.02 |
| | mIoU ↑ | 72.88 ± 0.44 | 73.14 ± 0.12 | 74.09 ± 0.28 | 75.55 ± 0.13 | 76.26 ± 0.17 | 76.68 ± 0.33 |
| | ECE ↓ | 0.68 ± 0.06 | 0.88 ± 0.01 | 0.98 ± 0.03 | 3.07 ± 0.01 | 2.49 ± 0.05 | 2.47 ± 0.01 |
| | BECE ↓ | 17.28 ± 0.19 | 13.25 ± 0.16 | 16.26 ± 0.18 | 32.10 ± 0.08 | 22.66 ± 0.10 | 22.57 ± 0.17 |
| DL3-MB2 | Acc ↑ | 95.24 ± 0.01 | 95.28 ± 0.03 | 95.30 ± 0.04 | 95.45 ± 0.02 | 95.51 ± 0.04 | 95.56 ± 0.03 |
| | mIoU ↑ | 72.19 ± 0.12 | 72.54 ± 0.26 | 72.88 ± 0.15 | 75.32 ± 0.42 | 75.81 ± 0.14 | 75.94 ± 0.24 |
| | ECE ↓ | 0.69 ± 0.05 | 0.96 ± 0.03 | 0.99 ± 0.04 | 3.09 ± 0.02 | 2.43 ± 0.04 | 2.50 ± 0.03 |
| | BECE ↓ | 17.64 ± 0.15 | 13.14 ± 0.20 | 16.55 ± 0.18 | 32.08 ± 0.29 | 21.61 ± 0.09 | 22.57 ± 0.04 |
| PSP-R18 | Acc ↑ | 95.13 ± 0.05 | 95.07 ± 0.03 | 95.14 ± 0.02 | 95.25 ± 0.02 | 95.30 ± 0.02 | 95.39 ± 0.01 |
| | mIoU ↑ | 72.61 ± 0.34 | 72.43 ± 0.05 | 72.79 ± 0.15 | 74.96 ± 0.16 | 75.31 ± 0.13 | 75.75 ± 0.31 |
| | ECE ↓ | 0.68 ± 0.07 | 0.97 ± 0.01 | 1.08 ± 0.05 | 3.17 ± 0.01 | 2.44 ± 0.02 | 2.52 ± 0.01 |
| | BECE ↓ | 17.54 ± 0.24 | 13.61 ± 0.20 | 16.96 ± 0.21 | 32.65 ± 0.07 | 22.30 ± 0.15 | 23.17 ± 0.08 |

Table 2: Results on PASCAL VOC (%). Best results within CE and JML groups are highlighted in red and green, respectively. Best results across CE and JML groups are underscored.

| Model | Metric | CE | CE-BLS | CE-KD | JML | JML-BLS | JML-KD |
|---|---|---|---|---|---|---|---|
| DL3-R101 | Acc ↑ | 94.68 ± 0.02 | 94.75 ± 0.05 | - | 94.97 ± 0.13 | 95.34 ± 0.08 | - |
| | mIoU ↑ | 78.39 ± 0.09 | 78.87 ± 0.44 | - | 80.26 ± 0.45 | 81.52 ± 0.41 | - |
| | ECE ↓ | 2.33 ± 0.03 | 2.01 ± 0.04 | - | 3.97 ± 0.10 | 3.20 ± 0.04 | - |
| | BECE ↓ | 20.54 ± 0.12 | 17.25 ± 0.14 | - | 32.70 ± 0.09 | 22.30 ± 0.02 | - |
| DL3-R50 | Acc ↑ | 94.23 ± 0.05 | 94.28 ± 0.06 | - | 94.60 ± 0.02 | 94.87 ± 0.02 | - |
| | mIoU ↑ | 76.93 ± 0.32 | 77.23 ± 0.18 | - | 78.97 ± 0.12 | 79.76 ± 0.15 | - |
| | ECE ↓ | 2.05 ± 0.03 | 1.77 ± 0.08 | - | 4.19 ± 0.02 | 3.25 ± 0.02 | - |
| | BECE ↓ | 20.95 ± 0.13 | 17.30 ± 0.15 | - | 33.07 ± 0.10 | 20.41 ± 0.01 | - |
| DL3-R18 | Acc ↑ | 92.96 ± 0.05 | 93.17 ± 0.09 | 93.13 ± 0.07 | 93.28 ± 0.14 | 93.72 ± 0.05 | 93.75 ± 0.05 |
| | mIoU ↑ | 72.47 ± 0.33 | 72.99 ± 0.42 | 72.96 ± 0.30 | 74.42 ± 0.52 | 75.60 ± 0.24 | 75.89 ± 0.29 |
| | ECE ↓ | 1.83 ± 0.12 | 1.26 ± 0.08 | 1.68 ± 0.05 | 4.92 ± 0.14 | 3.79 ± 0.03 | 3.94 ± 0.07 |
| | BECE ↓ | 21.83 ± 0.29 | 17.87 ± 0.08 | 19.42 ± 0.04 | 34.35 ± 0.09 | 22.45 ± 0.08 | 22.15 ± 0.19 |
| DL3-MB2 | Acc ↑ | 92.47 ± 0.12 | 92.45 ± 0.02 | 92.54 ± 0.04 | 92.74 ± 0.07 | 93.13 ± 0.11 | 93.21 ± 0.01 |
| | mIoU ↑ | 70.14 ± 0.54 | 70.36 ± 0.27 | 70.54 ± 0.10 | 72.35 ± 0.32 | 73.25 ± 0.18 | 73.55 ± 0.27 |
| | ECE ↓ | 1.92 ± 0.14 | 1.46 ± 0.09 | 1.78 ± 0.03 | 5.19 ± 0.05 | 4.14 ± 0.08 | 4.24 ± 0.01 |
| | BECE ↓ | 22.65 ± 0.34 | 19.62 ± 0.45 | 20.62 ± 0.34 | 35.10 ± 0.17 | 23.38 ± 0.19 | 23.09 ± 0.11 |
| PSP-R18 | Acc ↑ | 92.93 ± 0.12 | 93.09 ± 0.09 | 93.13 ± 0.18 | 93.20 ± 0.08 | 93.65 ± 0.04 | 93.60 ± 0.10 |
| | mIoU ↑ | 72.10 ± 0.55 | 72.76 ± 0.52 | 72.84 ± 0.57 | 74.20 ± 0.24 | 75.04 ± 0.22 | 74.93 ± 0.36 |
| | ECE ↓ | 1.92 ± 0.22 | 1.40 ± 0.16 | 1.81 ± 0.16 | 4.76 ± 0.04 | 3.70 ± 0.01 | 3.99 ± 0.07 |
| | BECE ↓ | 22.39 ± 0.39 | 18.95 ± 0.12 | 20.55 ± 0.47 | 34.72 ± 0.06 | 23.45 ± 0.20 | 23.53 ± 0.37 |

Table 3: Results on ADE20K (%). Best results within CE and JML groups are highlighted in red and green, respectively. Best results across CE and JML groups are underscored.

| Model | Metric | CE | CE-BLS | CE-KD | JML | JML-BLS | JML-KD |
|---|---|---|---|---|---|---|---|
| SegFormer-B5 | Acc ↑ | 82.48 ± 0.09 | 82.53 ± 0.06 | - | 82.72 ± 0.11 | 82.80 ± 0.10 | - |
| | mIoU ↑ | 48.17 ± 0.35 | 48.50 ± 0.26 | - | 49.95 ± 0.28 | 50.22 ± 0.32 | - |
| | ECE ↓ | 5.94 ± 0.05 | 5.14 ± 0.15 | - | 11.84 ± 0.06 | 9.12 ± 0.15 | - |
| | BECE ↓ | 19.98 ± 0.11 | 14.59 ± 0.17 | - | 34.78 ± 0.19 | 19.88 ± 0.28 | - |
| SegFormer-B3 | Acc ↑ | 81.91 ± 0.02 | 82.04 ± 0.11 | - | 82.31 ± 0.13 | 82.54 ± 0.12 | - |
| | mIoU ↑ | 46.51 ± 0.29 | 46.88 ± 0.24 | - | 48.68 ± 0.27 | 49.24 ± 0.24 | - |
| | ECE ↓ | 5.56 ± 0.12 | 4.56 ± 0.13 | - | 11.57 ± 0.17 | 10.63 ± 0.22 | - |
| | BECE ↓ | 19.98 ± 0.23 | 4.56 ± 0.13 | - | 34.47 ± 0.07 | 28.17 ± 0.36 | - |
| SegFormer-B1 | Acc ↑ | 78.75 ± 0.05 | 78.92 ± 0.11 | 78.96 ± 0.03 | 79.41 ± 0.10 | 79.53 ± 0.12 | 79.62 ± 0.04 |
| | mIoU ↑ | 38.79 ± 0.20 | 38.95 ± 0.25 | 39.29 ± 0.16 | 41.70 ± 0.17 | 42.03 ± 0.29 | 42.52 ± 0.17 |
| | ECE ↓ | 4.18 ± 0.04 | 3.18 ± 0.16 | 4.90 ± 0.05 | 11.33 ± 0.08 | 10.46 ± 0.18 | 10.50 ± 0.12 |
| | BECE ↓ | 18.70 ± 0.09 | 13.89 ± 0.20 | 19.23 ± 0.09 | 33.25 ± 0.05 | 27.38 ± 0.25 | 26.29 ± 0.33 |
| SegFormer-B0 | Acc ↑ | 75.49 ± 0.06 | 76.37 ± 0.02 | 76.48 ± 0.02 | 76.85 ± 0.04 | 76.98 ± 0.11 | 76.99 ± 0.06 |
| | mIoU ↑ | 30.49 ± 0.15 | 33.48 ± 0.07 | 34.54 ± 0.11 | 36.65 ± 0.13 | 36.78 ± 0.08 | 37.05 ± 0.11 |
| | ECE ↓ | 2.61 ± 0.12 | 2.10 ± 0.20 | 3.56 ± 0.04 | 10.77 ± 0.12 | 9.88 ± 0.17 | 10.22 ± 0.04 |
| | BECE ↓ | 17.61 ± 0.17 | 13.33 ± 0.12 | 17.53 ± 0.13 | 31.84 ± 0.28 | 26.09 ± 0.29 | 24.99 ± 0.14 |

Table 4: Comparing with SOTA segmentation KD methods on Cityscapes and PASCAL VOC. All results are mIoU (%). **JML-KD: we increase the batch size to 16 for Cityscapes experiments to match training details in** [70, 23, 24].

| Dataset | Model | CE | SKD [36] | IFVD [64] | CD [55] | CIRKD [70] | MasKD [24] | DIST [23] | JML-KD |
|---|---|---|---|---|---|---|---|---|---|
| CS | DL3-R18 | 72.88 | 75.42 | 75.59 | 75.55 | 76.38 | 77.00 | 77.10 | 77.91 ± 0.16 |
| | DL3-MB2 | 72.19 | 73.82 | 73.50 | 74.66 | 75.42 | 75.26 | - | 77.53 ± 0.20 |
| | PSP-R18 | 72.61 | 73.29 | 73.71 | 74.36 | 74.73 | 75.34 | 76.31 | 77.33 ± 0.38 |
| VOC | DL3-R18 | 72.47 | 73.51 | 73.85 | 74.02 | 74.50 | | | 75.89 ± 0.29 |
| | PSP-R18 | 72.10 | 74.07 | 73.54 | 73.99 | 74.78 | | | 74.93 ± 0.36 |

# 4 Ablation Studies

## 4.1 JML Weights

It is common in recent works [5, 9, 8, 29] to balance CE and the Dice loss with equal weights. For JML, we adopt 0.25/0.75 in all our experiments and find it slightly superior than 0.5/0.5.

Table 5: Ablating different values of $\lambda_{CE}/\lambda_{JML}$ on PASCAL VOC using DL3-R101 and DL3-R18. All results are mIoU (%). Red: the best in a row. Green: the worst in a row.

| Model | 0.10/0.90 | 0.25/0.75 | 0.50/0.50 | 0.90/0.10 | 1.00/0.00 |
|---|---|---|---|---|---|
| DL3-R101 | $79.80 \pm 0.40$ | $80.26 \pm 0.45$ | $79.92 \pm 0.12$ | $78.73 \pm 0.30$ | $78.39 \pm 0.09$ |
| DL3-R18 | $73.67 \pm 0.39$ | $74.42 \pm 0.52$ | $74.30 \pm 0.29$ | $73.21 \pm 0.23$ | $72.47 \pm 0.33$ |

## 4.2 JML-BLS

**Although BLS is sensitive to the choice of $\epsilon$, it effectively increases model accuracy and calibration.** The effect of the smoothing coefficient $\epsilon$ on PASCAL VOC with DL3-R101/50/18 is shown in Figure 5 (Appendix L). Interestingly, for DL3-R50 and DL3-R18, the optimal $\epsilon$ that achieves the highest mIoU also yields the lowest ECE.

**We need a strong smoothing coefficient near the boundary.** The optimal $\epsilon$ for different kernel size $k$ on PASCAL VOC with DL3-R18 is shown in Figure 4 (Appendix L). Note that $k = \infty$ implies LS is applied to every pixel, i.e. vanilla LS. Generally, as $k$ increases, we need to decrease the strength of smoothing. The best result is obtained when we only smooth a small region near the boundary with a strong smoothing coefficient ($k = 3$ and $\epsilon = 0.50$).

## 4.3 JML-KD

**Loss terms.** We examine the contribution of each loss term in Table 6 using a DL3-R18 student. Adding JML terms significantly improves the student's performance.

Table 6: Evaluating different losses terms on Cityscapes and PASCAL VOC using a DL3-R18 student. $\mathcal{L}_{JML\text{-}BLS}$ means we train the teacher with JML-BLS. All results are mIoU (%). Red: the best in a column.

| $\mathcal{L}_{CE, L}$ | $\mathcal{L}_{JML,L}$ | $\mathcal{L}_{CE, KD}$ | $\mathcal{L}_{JML, KD}$ | $\mathcal{L}_{JML\text{-}BLS}$ | CS | VOC |
|---|---|---|---|---|---|---|
| ✓ | - | - | - | - | $72.88 \pm 0.44$ | $72.47 \pm 0.33$ |
| ✓ | ✓ | - | - | - | $75.55 \pm 0.13$ | $74.42 \pm 0.52$ |
| ✓ | ✓ | ✓ | - | - | $75.74 \pm 0.25$ | $74.65 \pm 0.42$ |
| ✓ | ✓ | ✓ | ✓ | - | $76.16 \pm 0.37$ | $75.05 \pm 0.31$ |
| ✓ | ✓ | ✓ | ✓ | ✓ | $76.68 \pm 0.33$ | $75.89 \pm 0.29$ |

**Filtering out unimportant classes based on teacher's confidence is useful.** We examine the impact of active classes on PASCAL VOC with DL3-R18 in Table 7. In particular, we propose to ignore classes where the soft label, i.e. teacher's confidence, is low (marked as LABEL). And in ALL, we include all classes. In PRESENT, we select the class with the maximum confidence from the teacher's prediction in the class dimension. In PROB, we skip classes where the student itself is not confident. In BOTH, we take both the teacher's and student's confidences into account. The code to compute active classes is in Figure 6 (Appendix L).

We observe that using ALL classes in JML-KD can misguide the student. In most of the teacher's output classes, the confidence is usually very low. It can be challenging and potentially detrimental for the student to precisely mimic these numbers. With PRESENT, the performance remains similar to that without soft labels. This is because the effectiveness of using soft labels come from the non-argmax classes. PROB and BOTH achieve similar performance, but both are worse than LABEL.

Table 7: Comparing active classes on PASCAL VOC using DL3-R18. All results are mIoU (%).

| Active Classes | ALL | PRESENT | PROB | BOTH | LABEL (Ours) |
|---|---|---|---|---|---|
| JML-KD | $75.21 \pm 0.33$ | $74.50 \pm 0.25$ | $75.51 \pm 0.53$ | $75.39 \pm 0.31$ | $75.89 \pm 0.29$ |

**Teacher's calibration is beneficial.** In Sec. 2.4.2, we suggest that a teacher trained with JML-BLS offers more boundary information to the student. But what exactly is this boundary information? We

believe it relates to the teacher's calibration. The student, with less capacity, often struggles to mimic a more powerful teacher, especially near ambiguous boundary regions. However, if the teacher is more calibrated and outputs a less peaked distribution, the student will learn this uncertainty rather than attempting to match an unrealistic distribution that it lacks the capacity to replicate.

In Table 8, we compare three teachers with various accuracy and calibration on PASCAL VOC. In particular, both T2 and T3 are trained with JML-BLS but with different smoothing parameter $\epsilon$, while T1 is not. T2, although having a lower mIoU, is more calibrated than T1. Consequently, T2's student is more accurate and better calibrated than T1's.

Table 8: Comparing 3 teachers of different accuracy and calibration on PASCAL VOC (%). Prefix T stands for the teacher and prefix S the student. Red: the best in a row.

| Metrics | T1 | T2 | T3 |
|---|---|---|---|
| T ECE ↓ | 4.20 | 3.47 | 3.24 |
| T BECE ↓ | 33.17 | 20.73 | 20.42 |
| T mIoU ↑ | 79.09 | 78.75 | 79.82 |
| S ECE ↓ | $4.76 \pm 0.09$ | $4.01 \pm 0.07$ | $3.94 \pm 0.07$ |
| S BECE ↓ | $33.86 \pm 0.10$ | $22.21 \pm 0.06$ | $22.15 \pm 0.05$ |
| S mIoU ↑ | $75.05 \pm 0.31$ | $75.43 \pm 0.35$ | $75.89 \pm 0.29$ |

## 5   Related Works

IoU is a commonly employed metric in semantic segmentation, and IoU losses strive to optimize this metric directly. IoU only obtains values when both predictions and ground-truth labels are discrete binary vectors $\{0, 1\}^p$, but the neural network often predicts soft probabilities (after the softmax or sigmoid layer) in $[0, 1]^p$. Interpolating IoU values from $\{0, 1\}^p$ to $[0, 1]^p$ has primarily followed two routes. One involves relaxing set counting as norm functions, as seen in the soft Jaccard loss [46, 49], the soft Dice loss [57], the soft Tversky loss [53] and the focal Tversky loss [1]. The other capitalizes on the fact that IoU is submodular [72], allowing for the application of the convex Lovasz extension of submodular functions. For instance, the Lovasz-Softmax loss [2], the Lovasz hinge loss [72] and the PixIoU loss [73]. Nevertheless, as IoU values are extended for predictions from $\{0, 1\}^p$ to $[0, 1]^p$, the fact that labels can also fall within $[0, 1]^p$ is often overlooked. As a result, these IoU losses are incompatible with soft labels.

Besides semantic segmentation, IoU is adopted across a wide spectrum of fields. Due to its discrete nature, its probabilistic extensions [56] have found use in object detection [19], medical imaging [16] and information retrieval [26, 43].

## 6   Discussion

### 6.1   How to tune the hyper-parameters of JML?

In this section, we delve into some critical hyper-parameters of JML. For a comprehensive list, please refer to the accompanying code.

**mIoU$^{\text{D}}$/mIoU$^{\text{I}}$/mIoU$^{\text{C}}$** (default: 1.0/0.0/0.0): the weight of the loss to optimize mIoU$^{\text{D}}$/mIoU$^{\text{I}}$/mIoU$^{\text{C}}$ [65]. The appropriate choice is mainly dependent on the targeted evaluation metrics. Given that the prevailing metric is mIoU$^{\text{D}}$ (per-dataset mIoU), we recommend to set mIoU$^{\text{D}}$ to 1.0 and mIoU$^{\text{I}}$/mIoU$^{\text{C}}$ to 0.0. However, it is imperative to acknowledge the inherent trade-offs when using JML to optimize different metrics [65].

**alpha/beta** (default: 1.0/1.0): the coefficient of false positives/negatives in the Tversky loss. For instance, when `alpha` and `beta` are both set to 1.0, the configuration corresponds to JML. Conversely, an `alpha` and `beta` value of 0.5 each leads to DML [66]. When the evaluation metric is IoU, JML is advised, while it is the Dice score, DML is more appropriate. The general Tversky loss is useful when separate weights are required for false positives/negatives.

**active_classes_mode** (default: `PRESENT` for hard labels and `ALL` for soft labels): the mode to compute active classes. With hard labels, it is suggested to use `PRESENT`, since the loss aligns more effectively with the evaluation metric, especially when (i) the dataset contains a large number of classes (e.g. ADE20K), and/or (ii) evaluated with fine-grained mIoUs [65]. With soft labels, the optimal choice varies based on specific applications (see Table 7).

### 6.2 How to use JML?

**Combine JML with CE.** Our findings suggest that incorporating CE, particularly during the initial stage of training, can expedite convergence (as illustrated in Figure 1). Consequently, relying solely on JML is not recommended. For a discussion on the balancing weights, please consult Section 4.1.

**Tune the hyper-parameters of JML.** We have endeavored to minimize the necessity of extensive tuning. In the majority of scenarios, default hyper-parameter values (as illustrated above) should suffice. However, some situations might benefit from further refinement. In particular, pay attention to `active_classes_mode` when dealing with soft labels.

**Tune the hyper-parameters of training settings.** Current training recipes have been heavily optimized for CE. Although we find JML generally aligns with these hyper-parameters, due diligence is required. Notably, one of the advantages of JML is its ability to speed up convergence in the later training phase (as depicted in Figure 1). As a consequence, training with a combination of CE and JML often converges in considerably fewer epochs compared to training with CE alone. Please refer to [65] for a comparison of the number of epochs with those in MMSegmentation.

**Be careful with distributed training.** In our concurrent work [65], we observe that the loss presents trade-offs when optimizing different mIoU variants. If the per-GPU batch size is small (which is often the case) and the loss is computed on each GPU independently, the loss may inadvertently optimize for mIoU$^\text{I}$ (per-image mIoU). This can lead to suboptimal outcomes when evaluated with mIoU$^\text{D}$ (per-dataset mIoU).

**Be careful with extremely class-imbalance.** Aligned with mIoU, JML is designed to compute the loss on a per-class basis and subsequently average these individual losses. In preliminary experiments on some medical datasets, we identified instances of severe class imbalances. In such scenarios, the class-wise loss may inadvertently amplify the significance of underrepresented classes, potentially disturb the training process. To address this, one might consider (i) oversampling the minority classes, and/or (ii) adopting a class-agnostic loss computation, where the intersection and union are calculated over all pixels.

## 7 Limitation

In both this study and our subsequent work [66], the focus is to extend the losses in the field of image segmentation. It would be intriguing to apply these losses to other tasks, such as long-tailed classification [34]. Moreover, although we adopt these losses in the label space, they present potential in quantifying the similarity between two feature vectors [24], potentially serving as an alternative to the $L^p$ norm or cosine similarity.

## 8 Conclusion

This paper is driven by the observation that current IoU losses fall short when dealing with soft labels, which substantially limits their adaptability to crucial training techniques. To address this limitation, we introduce the Jaccard metric losses (JMLs). While these losses are identical to the soft Jaccard loss in a conventional hard-label setting, they offer full compatibility with soft labels.

Our results demonstrate that integrating JMLs with label smoothing, knowledge distillation, and semi-supervised learning leads to notable improvements in both accuracy and calibration. This is consistent across a spectrum of datasets and network architectures, encompassing classic CNNs as well as recent vision transformers. Remarkably, the proposed methods, which are simple and solely rely on soft labels, surpass state-of-the-art segmentation knowledge distillation and semi-supervised learning techniques by a significant margin.

In our follow-up study [66], we delve into the extensions of various other losses, including the soft Dice loss, soft Tversky loss, and focal Tversky loss. Given their equivalence to their original counterparts in a standard hard-label context and their enhanced compatibility with soft labels, we recommend to replace the existing implementations with ours.

## Acknowledgements

We acknowledge support from the Research Foundation - Flanders (FWO) through project numbers G0A1319N and S001421N, and funding from the Flemish Government under the Onderzoeksprogramma Artificiële Intelligentie (AI) Vlaanderen programme. The resources and services used in this work were provided by the VSC (Flemish Supercomputer Center), funded by the Research Foundation - Flanders (FWO) and the Flemish Government.

We thank Ilya Bogdanov for his help on the proof.

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

## A  Architectures

Our study includes 13 models, comprising both classic CNNs and recent vision transformers. Details of each model are presented in Table 9. FLOPs computations are based on an input size of $512 \times 1024$. Inference latency measurements are conducted with the same input size on a NVIDIA A100. We estimate the training memory requirements using a ground-truth size of $8 \times 19 \times 512 \times 1024$ (`batch_size`, `num_classes`, `H`, `W`), also on a NVIDIA A100.

Table 9: Details of each model.

| Method | Backbone | #Params (M) | FLOPs (G) | Latency (ms) | Memory (GB) |
|---|---|---|---|---|---|
| DeepLabV3+ [7] | ResNet101 [21] | 62.58 | 528.32 | 26.47 | 42.68 |
| DeepLabV3 [6] | ResNet101 [21] | 87.10 | 714.73 | 26.20 | 33.99 |
| DeepLabV3 [6] | ResNet50 [21] | 68.11 | 559.14 | 17.48 | 21.02 |
| DeepLabV3 [6] | ResNet18 [21] | 13.98 | 121.29 | 4.93 | 6.43 |
| DeepLabV3 [6] | MobileNetV2 [54] | 3.79 | 31.68 | 5.89 | 16.74 |
| PSPNet [6] | ResNet18 [21] | 12.79 | 109.90 | 4.82 | 6.26 |
| UNet [51] | ResNet50 [21] | 76.07 | 409.35 | 18.09 | 24.76 |
| UNet [51] | ResNet18 [21] | 14.49 | 46.42 | 6.15 | 7.76 |
| UNet [51] | MobileNetV2 [54] | 4.06 | 22.63 | 8.15 | 11.16 |
| SegFormer [69] | MiTB5 [69] | 83.77 | 198.37 | 43.32 | 41.99 |
| SegFormer [69] | MiTB3 [69] | 46.40 | 124.52 | 26.15 | 28.29 |
| SegFormer [69] | MiTB1 [69] | 15.48 | 63.85 | 11.07 | 15.99 |
| SegFormer [69] | MiTB0 [69] | 5.01 | 45.46 | 8.98 | 13.45 |

## B  Training Details

By default, we adopted the training details outlined in [70, 23, 24], except for the reduction of the batch size to 8. In particular, we utilize SGD with a weight decay of 0.0005 and a momentum of 0.9. The initial learning rate is 0.01, and is decayed according to $(1 - \frac{\text{iter}}{\text{total iters}})^{0.9}$. The number of iterations is 40K for Cityscapes [11] and PASCAL VOC [18], 10K for DeepGlobe Land [12]. The crop size is $512 \times 1024$ for Cityscapes [11], $512 \times 512$ for PASCAL VOC [18] and DeepGlobe Land [12]. In Table 4, for a fair comparison with the baselines, we increase the batch size of JML-KD to 16 in Cityscapes experiments, and all other parameters retain at their default values.

For experiments with vision transformers, we adhered to the training recipes in [10]. Specifically, we use AdamW [38] with a weight decay of 0.01. We begin with an initial learning rate of 0.00006, and is decayed by $(1 - \frac{\text{iter}}{\text{total iters}})^{1}$. The number of iterations is 40K and the crop size is $512 \times 512$ for ADE20K [81]. The batch size is 8.

For semi-supervised learning, we follow training details in [80]. In particular, we use SGD with a weight decay of 0.0001 and a momentum of 0.9. The initial learning rate is 0.001, and is decayed according to $(1 - \frac{\text{iter}}{\text{total iters}})^{0.9}$. The number of epochs is 80 and the crop size is $512 \times 512$ for PASCAL VOC [18]. The batch size is 16.

For JML experiments, we set $\mu_L/\mu_T/\nu_L/\nu_T/\eta_S/\eta_U/\theta_S/\theta_U$ to 0.5 and $\lambda_{\text{CE}}/\lambda_{\text{JML}}$ to 0.25/0.75.

## C  More analysis of $\overline{\Delta}_{\text{SJL},L^1}$

For $x, y \in [0,1]^p$, let us rewrite $\overline{\Delta}_{\text{SJL},L^1}$ for a particular pixel $i$:

$$\overline{\Delta}_{\text{SJL},L^1} = 1 - \frac{x_i y_i + b}{x_i + y_i - x_i y_i + a} \tag{14}$$

such that

$$a = \sum_{j \neq i} x_j + \sum_{j \neq i} y_j - \sum_{j \neq i} x_j y_j, \tag{15}$$

$$b = \sum_{j \neq i} x_j y_j. \tag{16}$$

Assume $ab \neq 0$ and predictions are independent from each other. Take the derivative with respect to $x_i$:

$$\frac{\mathrm{d}\overline{\Delta}_{\text{SJL},L^1}}{\mathrm{d}x_i} = -\frac{y_i^2 + (a+b)y_i - b}{(x_i + y_i - x_i y_i + a)^2}. \tag{17}$$

The numerator has two roots:

$$r_1 = \frac{-(a+b) - \sqrt{(a+b)^2 + 4b}}{2}, \tag{18}$$

$$r_2 = \frac{-(a+b) + \sqrt{(a+b)^2 + 4b}}{2}. \tag{19}$$

It is easy to see that $r_1 \leq 0 \leq r_2 \leq 1$. Therefore we have

$$\frac{\mathrm{d}\overline{\Delta}_{\mathrm{SJL},L^1}}{\mathrm{d}x_i} \begin{cases} \leq 0 & \text{if } y_i \geq r_2, \\ > 0 & \text{otherwise.} \end{cases} \tag{20}$$

That is, $\overline{\Delta}_{\mathrm{SJL},L^1}$ will push $x_i$ towards either 1 if $y_i > r_2$, or 0 if $y_i < r_2$, rather than making it close to $y_i$. In practice, predictions are not independent from each other, hence the exact behavior of $\overline{\Delta}_{\mathrm{SJL},L^1}$ is a complex process. However, the above analysis still provides an insight of how $\overline{\Delta}_{\mathrm{SJL},L^1}$ will react to soft labels.

## D   The Lovasz-Softmax Loss

Based on the fact that IoU is submodular [72], the Lovasz-Softmax loss (LSL) [2] computes the convex Lovasz extension of IoU. Specifically, for each prediction $x_i \in [0, 1]$, mispredictions are computed as

$$m_i = \begin{cases} 1 - x_i & \text{if } y_i = 1, \\ x_i & \text{otherwise.} \end{cases} \tag{21}$$

The Lovasz extension can be applied such that

$$\overline{\Delta}_{\mathrm{LSL}} : m \in [0,1]^p, y \in \{0,1\}^p \mapsto \sum_{i=1}^{p} m_i g_i(m)$$

where $g_i(m) = \Delta_{\mathrm{IoU}}(\{\pi_1, ..., \pi_i\}) - \Delta_{\mathrm{IoU}}(\{\pi_1, ..., \pi_{i-1}\})$ and $\pi$ is a permutation ordering of $m$ such that $m_{\pi_1} \geq ... \geq m_{\pi_p}$.

Loss functions that rely on the Lovasz extension, such as LSL [2], the Lovasz hinge loss [72] and the PixIoU loss [73] cannot handle soft labels because they need to compute $g_i(m)$ and it requires $y \in \{0,1\}^p$.

## E   JMLs vs. other IoU Losses

In Table 10, we evaluate the performance of $\overline{\Delta}_{\mathrm{JML},1}, \overline{\Delta}_{\mathrm{JML},2}, \overline{\Delta}_{\mathrm{SJL},L^1}, \overline{\Delta}_{\mathrm{SJL},L^2}$ and $\overline{\Delta}_{\mathrm{LSL}}$ on Cityscapes and PASCAL VOC using DL3-R18.

As elaborated in Appendix C, $\overline{\Delta}_{\mathrm{SJL},L^1}$ is equivalent to $\overline{\Delta}_{\mathrm{JML}}$ in the context of hard labels. Nevertheless, with soft labels, $\overline{\Delta}_{\mathrm{SJL},L^1}$ has a tendency to push predictions towards vertices, rather than optimizing for the ideal $x = y$ scenario. Our experiments on PASCAL VOC reveal that models trained with $\overline{\Delta}_{\mathrm{SJL},L^1}$ using soft labels are outperformed by their hard-label counterparts. In fact, predictions for multiple classes might be simultaneously pushed towards 1, creating a conflict and potentially destabilizing the training process.

Conversely, $\overline{\Delta}_{\mathrm{SJL},L^2}$ is free from the aforementioned issue, as highlighted in Section 2.2. Nevertheless, it is found to be less effective than $\overline{\Delta}_{\mathrm{SJL},L^1}$ when employed with hard labels, possibly due to its flattened behavior near the minimum [17]. Our empirical results further substantiate that it is surpassed by $\overline{\Delta}_{\mathrm{JML}}$ on Cityscapes and PASCAL VOC, for both hard and soft labels.

Lastly, $\overline{\Delta}_{\mathrm{LSL}}$ cannot take soft labels as input, given its reliance on the Lovasz extension. Our evaluations indicate that its performance is similar to that of $\overline{\Delta}_{\mathrm{JML}}$ with hard labels.

Table 10: Comparing $\overline{\Delta}_{\mathrm{JML},1}, \overline{\Delta}_{\mathrm{JML},2}, \overline{\Delta}_{\mathrm{SJL},L^1}, \overline{\Delta}_{\mathrm{SJL},L^2}$ and $\overline{\Delta}_{\mathrm{LSL}}$ on Cityscapes and PASCAL VOC using DL3-R18. All results are mIoU (%). Red: the best in a column.

| Dataset | Loss | Hard | BLS | KD |
|---|---|---|---|---|
| CS | $\overline{\Delta}_{\mathrm{JML},1}$ | $75.55 \pm 0.13$ | $76.26 \pm 0.17$ | $76.68 \pm 0.33$ |
| | $\overline{\Delta}_{\mathrm{JML},2}$ | $75.55 \pm 0.13$ | $76.31 \pm 0.23$ | $76.45 \pm 0.26$ |
| | $\overline{\Delta}_{\mathrm{SJL},L^1}$ | $75.55 \pm 0.13$ | $75.78 \pm 0.34$ | $75.83 \pm 0.43$ |
| | $\overline{\Delta}_{\mathrm{SJL},L^2}$ | $75.28 \pm 0.26$ | $75.51 \pm 0.38$ | $75.87 \pm 0.52$ |
| | $\overline{\Delta}_{\mathrm{LSL}}$ | $75.53 \pm 0.26$ | - | - |
| VOC | $\overline{\Delta}_{\mathrm{JML},1}$ | $74.42 \pm 0.52$ | $75.60 \pm 0.24$ | $75.89 \pm 0.29$ |
| | $\overline{\Delta}_{\mathrm{JML},2}$ | $74.42 \pm 0.52$ | $75.31 \pm 0.31$ | $75.49 \pm 0.35$ |
| | $\overline{\Delta}_{\mathrm{SJL},L^1}$ | $74.42 \pm 0.52$ | $74.13 \pm 0.20$ | $74.24 \pm 0.61$ |
| | $\overline{\Delta}_{\mathrm{SJL},L^2}$ | $73.72 \pm 0.30$ | $74.87 \pm 0.38$ | $74.83 \pm 0.32$ |
| | $\overline{\Delta}_{\mathrm{LSL}}$ | $74.45 \pm 0.36$ | - | - |

# F    Proof of Theorem 2.1

*Proof.*     (i) $\overline{\Delta}_{\mathrm{JML},1}$ is a metric on $[0,1]^p$. The proof was given in [56]. Here we provide a sketch for the proof of the triangle inequality. Recall the definition of $\overline{\Delta}_{\mathrm{JML},1}$:

$$\overline{\Delta}_{\mathrm{JML},1} = 1 - \frac{\|x\|_1 + \|y\|_1 - \|x-y\|_1}{\|x\|_1 + \|y\|_1 + \|x-y\|_1} = \frac{2\|x-y\|_1}{\|x\|_1 + \|y\|_1 + \|x-y\|_1}. \tag{22}$$

Given a fixed point $a \in [0,1]^p$, the key is to define a new function $d'$:

$$d'(x,y) = \begin{cases} 0 & \text{if } x = y = a, \\ \frac{d(x,y)}{d(x,a)+d(y,a)+d(x,y)} & \text{otherwise;} \end{cases} \tag{23}$$

and show that if $d$ is a metric on $[0,1]^p$, then $d'$ is also a metric on $[0,1]^p$. The proof can be found in [56].

Substitute the definition of the $L^1$ norm as $d$ and let $a = 0$. Since the $L^1$ norm is a metric on $[0,1]^p$, we can conclude that $\overline{\Delta}_{\mathrm{JML},1}$ is also a metric on $[0,1]^p$.

(ii) $\overline{\Delta}_{\mathrm{JML},2}$ is a metric on $[0,1]^p$. Conditions (i) - (iii) are obvious. To show the triangle inequality, note that

$$\langle b, c \rangle = \sum_i b_i c_i \tag{24}$$

$$= \sum_i \left( (b_i - a_i)c_i + a_i c_i \right) \tag{25}$$

$$\leq \sum_i |b_i - a_i| + \sum_i a_i c_i \tag{26}$$

$$= \|a-b\|_1 + \langle a, c \rangle. \tag{27}$$

Similarly,

$$\langle a, b \rangle \leq \|b-c\|_1 + \langle a, c \rangle. \tag{28}$$

Hence

$$\overline{\Delta}_{\mathrm{JML},2}(a,b) + \overline{\Delta}_{\mathrm{JML},2}(b,c) \tag{29}$$

$$= \frac{\|a-b\|_1}{\|a-b\|_1 + \langle a,b \rangle} + \frac{\|b-c\|_1}{\|b-c\|_1 + \langle b,c \rangle} \tag{30}$$

$$\geq \frac{\|a-b\|_1}{\|a-b\|_1 + \|b-c\|_1 + \langle a,c \rangle} + \frac{\|b-c\|_1}{\|a-b\|_1 + \|b-c\|_1 + \langle a,c \rangle} \tag{31}$$

$$= \frac{\|a-b\|_1 + \|b-c\|_1}{\|a-b\|_1 + \|b-c\|_1 + \langle a,c \rangle} \tag{32}$$

$$\geq \frac{\|a-c\|_1}{\|a-c\|_1 + \langle a,c \rangle} \tag{33}$$

$$= \overline{\Delta}_{\mathrm{JML},2}(a,c). \tag{34}$$

Note that if $p \geq q > 0, r \geq 0$, we have $\frac{p}{p+r} \geq \frac{q}{q+r}$. The last inequality follows from $\|a-b\|_1 + \|b-c\|_1 \geq \|a-c\|_1$.

(iii) $\overline{\Delta}_{\text{SJL},L^1}$ is not a metric on $[0,1]^p$. For instance, if $a = 0.5$, $\overline{\Delta}_{\text{SJL},L^1}(a,a) = 2/3 \neq 0$.

(iv) $\overline{\Delta}_{\text{SJL},L^2}$ is not a metric on $[0,1]^p$. For instance, if $a = 0.8, b = 0.4, c = 0.2$, it does not satisfy the triangle inequality.

□

## G  Proof of Theorem 2.3

*Proof.*    (i) $\forall x \in [0,1]^p$, $y \in \{0,1\}^p$ and $x \in \{0,1\}^p$, $y \in [0,1]^p$, $\overline{\Delta}_{\text{SJL},L^1} = \overline{\Delta}_{\text{JML},1} = \overline{\Delta}_{\text{JML},2}$. Due to symmetry, we only need to prove the first part. Note that

$$\|x\|_1 = \sum_i x_i = \sum_i \mathbb{1}_{(y_i=0)} x_i + \sum_i \mathbb{1}_{(y_i=1)} x_i \tag{35}$$

$$\|y\|_1 = \sum_i y_i = \sum_i \mathbb{1}_{(y_i=1)} \tag{36}$$

$$\langle x, y \rangle = \sum_i x_i y_i = \sum_i \mathbb{1}_{(y_i=1)} x_i \tag{37}$$

$$\|x - y\|_1 = \sum_i |x_i - y_i| = \sum_i \mathbb{1}_{(y_i=0)} x_i - \sum_i \mathbb{1}_{(y_i=1)} x_i + \sum_i \mathbb{1}_{(y_i=1)}. \tag{38}$$

Combine these, we have

$$\overline{\Delta}_{\text{SJL},L^1} = \overline{\Delta}_{\text{JML},1} = \overline{\Delta}_{\text{JML},2} = \frac{\sum_i \mathbb{1}_{(y_i=1)} x_i}{\sum_i \mathbb{1}_{(y_i=0)} x_i + \sum_i \mathbb{1}_{(y_i=1)}}. \tag{39}$$

(ii) $\exists x, y \in [0,1]^p, \overline{\Delta}_{\text{SJL},L^1} \neq \overline{\Delta}_{\text{JML},1} \neq \overline{\Delta}_{\text{JML},2}$. For instance, if $x = 0.8, y = 0.5, \overline{\Delta}_{\text{SJL},L^1} \neq \overline{\Delta}_{\text{JML},1} \neq \overline{\Delta}_{\text{JML},2}$.

□

## H  $\overline{\Delta}_{\text{JML,1}}$ vs. $\overline{\Delta}_{\text{JML,2}}$

**Definition H.1** (Convex Closure). The convex closure of a set function $f : x \in \{0,1\}^p \to [0,1]$ is

$$\mathbb{C}f : [0,1]^p \to [0,1] = \min_{\alpha} \sum_{i=1}^p \alpha_i f(x_i), \tag{40}$$

$$\text{s.t. } \sum_{i=1}^p \alpha_i = 1, \tag{41}$$

$$\sum_{i=1}^p \alpha_i x_i = x, \tag{42}$$

$$\alpha_i \geq 0. \tag{43}$$

The convex closure extends a set function that is only defined at the vertices $\{0,1\}^p$ to the whole hypercube $[0,1]^p$ by linearly interpolating the values at these vertices. We plot the loss value of $\overline{\Delta}_{\text{JML},1}, \overline{\Delta}_{\text{JML},2}$ and the convex closure of $\overline{\Delta}_{\text{SJL},L^1}$ in Figure 2 when $y = 0.5$. Note that $\overline{\Delta}_{\text{JML},1}$ overlaps with the convex closure at $[0, 0.5]$. We can see that $\overline{\Delta}_{\text{JML},1} \leq \overline{\Delta}_{\text{JML},2}$ and $\overline{\Delta}_{\text{JML},1}$ is closer to the convex closure. Generally, we have

**Theorem H.2.** $\forall x, y \in [0,1]^p, \overline{\Delta}_{\text{JML,1}} \leq \overline{\Delta}_{\text{JML,2}}$.

*Proof.*

$$\overline{\Delta}_{\text{JML,1}} \leq \overline{\Delta}_{\text{JML,2}} \tag{44}$$

$$\Rightarrow \frac{\|x\|_1 + \|y\|_1 - \|x - y\|_1}{\|x\|_1 + \|y\|_1 + \|x - y\|_1} - \frac{\langle x, y \rangle}{\langle x, y \rangle + \|x - y\|_1} \geq 0 \tag{45}$$

$$\Rightarrow (\|x\|_1 + \|y\|_1)\|x - y\|_1 - \|x - y\|_1^2 - 2\langle x, y \rangle\|x - y\|_1 \geq 0. \tag{46}$$

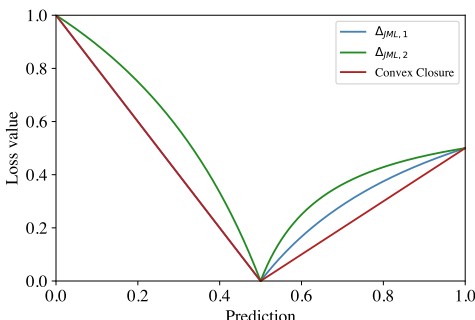

Figure 2: Comparing $\overline{\Delta}_{\text{JML},1}, \overline{\Delta}_{\text{JML},2}$ and the convex closure with $y = 0.5$.

We can switch the element of $x$ and $y$ whenever $x_i \leq y_i$ for some $i$. Doing this, each term in the last inequality remains the same. Let us denote the new variable as $x', y'$ such that $x'_i \geq y'_i$ for all $i$. Hence

$$(\|x\|_1 + \|y\|_1)\|x - y\|_1 - \|x - y\|_1^2 - 2\langle x, y \rangle \|x - y\|_1 \tag{47}$$

$$= (\|x'\|_1 + \|y'\|_1)\|x' - y'\|_1 - \|x' - y'\|_1^2 - 2\langle x', y' \rangle \|x' - y'\|_1 \tag{48}$$

$$= (\|x'\|_1 + \|y'\|_1)(\|x'\|_1 - \|y'\|_1) - (\|x'\|_1 - \|y'\|_1)^2 - 2\langle x', y' \rangle (\|x'\|_1 - \|y'\|_1) \tag{49}$$

$$= 2(\|x'\|_1 - \|y'\|_1)(\|y'\|_1 - \langle x', y' \rangle) \tag{50}$$

$$= 2 \sum_i (x'_i - y'_i) \sum_i (1 - x'_i)y'_i \geq 0. \tag{51}$$

$\square$

**Theorem H.3.** *Given a set function $l : \{0,1\}^p \to [0,1]$ and two concave functions $f, g : [0,1]^p \to [0,1]$. If $\forall x \in \{0,1\}^p, l(x) = f(x) = g(x)$ and $\forall x \in [0,1]^p, f(x) \leq g(x)$, then $\forall x \in [0,1]^p, |f(x) - \mathbb{C}l(x)| \leq |g(x) - \mathbb{C}l(x)|$.*

*Proof.* It suffices to show $\forall x \in [0,1]^p, f(x) \geq \mathbb{C}l(x)$. $\forall x \in [0,1]^p$, we have $x = \sum_{i=1}^p \alpha_i x_i$ such that $\alpha_i \geq 0, \sum_{i=1}^p \alpha_i = 1$ and $x_i \in \{0,1\}$. Therefore

$$f(x) = f(\sum_{i=1}^p \alpha_i x_i) \tag{52}$$

$$= f\left((1 - \alpha_p)\frac{\sum_{i=1}^{p-1} \alpha_i x_i}{1 - \alpha_p} + \alpha_p x_p\right) \tag{53}$$

$$\geq (1 - \alpha_p)f\left(\frac{\sum_{i=1}^{p-1} \alpha_i x_i}{1 - \alpha_p}\right) + \alpha_p f(x_p) \tag{54}$$

$$= (1 - \alpha_p)f\left(\frac{1 - \alpha_p - \alpha_{p-1}}{1 - \alpha_p}\frac{\sum_{i=1}^{p-2} \alpha_i x_i}{1 - \alpha_p - \alpha_{p-1}} + \frac{\alpha_{p-1}}{1 - \alpha_p}x_{p-1}\right) + \alpha_p f(x_p) \tag{55}$$

$$\geq (1 - \alpha_p - \alpha_{p-1})f\left(\frac{\sum_{i=1}^{p-2} \alpha_i x_i}{1 - \alpha_p - \alpha_{p-1}}\right) + \alpha_{p-1}f(x_{p-1}) + \alpha_p f(x_p) \tag{56}$$

$$\geq ... \geq \sum_{i=1}^p \alpha_i f(x_i) \tag{57}$$

$$= \sum_{i=1}^p \alpha_i l(x_i) \tag{58}$$

$$\geq \mathbb{C}l(x). \tag{59}$$

$\square$

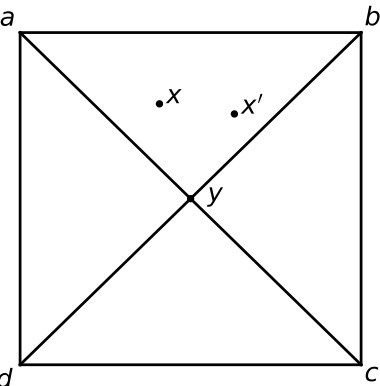

Figure 3: A counterexample that both $\overline{\Delta}_{\text{JML,1}}$ and $\overline{\Delta}_{\text{JML,2}}$ are not piece-wise concave in 2D.

Although both $\overline{\Delta}_{\text{JML,1}}$ and $\overline{\Delta}_{\text{JML,2}}$ are not concave as Figure 2 shows, if we can divide the hypercube into several sub-spaces such that $\overline{\Delta}_{\text{JML,1}}$ and $\overline{\Delta}_{\text{JML,2}}$ are equal at the vertices and concave at each sub-space, then we can still apply Theorem H.3. However, the fact that both of them are piece-wise concave in 1D does not hold in higher dimension. Indeed, we can find a counterexample in 2D. Let $y = [0.5, 0.5], x = [0.4087, 0.7855], x' = [0.6285, 0.7551]$. Both $x$ and $x'$ are in the sub-space $yab$ and

$$0.5\overline{\Delta}_{\text{JML,1}}(x, y) + 0.5\overline{\Delta}_{\text{JML,1}}(x', y) > \overline{\Delta}_{\text{JML,1}}(0.5x + 0.5x', y), \tag{60}$$

$$0.5\overline{\Delta}_{\text{JML,2}}(x, y) + 0.5\overline{\Delta}_{\text{JML,2}}(x', y) > \overline{\Delta}_{\text{JML,2}}(0.5x + 0.5x', y). \tag{61}$$

How do they perform empirically? In Table 10, we compare $\overline{\Delta}_{\text{JML,1}}$ and $\overline{\Delta}_{\text{JML,2}}$ on Cityscapes and PASCAL VOC using DL3-R18. We find $\overline{\Delta}_{\text{JML,1}}$ is slightly superior to $\overline{\Delta}_{\text{JML,2}}$.

## I  Semi-supervised Learning

We focus on augmentation-based semi-supervised learning approaches [4, 80]. Broadly speaking, in their approaches, the data consists of both supervised and unsupervised images, and a model $H$ is trained to minimize the following loss:

$$\mathcal{L}_{\text{CE, SSL}} = \eta_{\text{S}}\mathcal{L}_{\text{CE, S}} + \eta_{\text{U}}\mathcal{L}_{\text{CE, U}} \tag{62}$$

such that $\mathcal{L}_{\text{CE, S}} = \text{CE}(H(\mathcal{A}_s(x_s)), L(\mathcal{A}_s(x_s)))$ and $\mathcal{L}_{\text{CE, U}} = \text{CE}(H(\mathcal{A}_u(x_u)), T(\mathcal{A}_u(x_u)))$.

Here, $\mathcal{A}_s$ and $\mathcal{A}_u$ are data augmentations applied to labeled images $x_s$ and unlabeled images $x_u$, respectively. $L$ is the ground-truth label and $T$ is usually the exponential moving averaging of the model weights.

In JML-SSL, we introduce additional supervision from JML:

$$\mathcal{L}_{\text{JML, SSL}} = \theta_{\text{S}}\mathcal{L}_{\text{JML, S}} + \theta_{\text{U}}\mathcal{L}_{\text{JML, U}} \tag{63}$$

where $\mathcal{L}_{\text{JML, S}} = \text{JML}(H(\mathcal{A}_s(x_s)), L(\mathcal{A}_s(x_s)))$ and $\mathcal{L}_{\text{JML, U}} = \text{JML}(H(\mathcal{A}_u(x_u)), T(\mathcal{A}_u(x_u)))$.

Combining these yields

$$\mathcal{L}_{\text{JML-SSL}} = \lambda_{\text{CE}}\mathcal{L}_{\text{CE, SSL}} + \lambda_{\text{JML}}\mathcal{L}_{\text{JML, SSL}}. \tag{64}$$

To further harness the power of soft labels, we follow the pipelines of AugSeg [80] and stack BLS on top of their augmentations: $\mathcal{A}_s(x_s) = \text{BLS}(\mathcal{A}_{s,\text{AugSeg}}(x_s))$ and $\mathcal{A}_u(x_u) = \text{BLS}(\mathcal{A}_{u,\text{AugSeg}}(x_u))$.

Here, $\mathcal{A}_{s,\text{AugSeg}}$ and $\mathcal{A}_{u,\text{AugSeg}}$ denote augmentations applied to labeled and unlabeled images in AugSeg, respectively. We refer readers to their paper for more details.

As shown in Table 11, we achieve SOTA results on semi-supervised learning. In Table 12, we ablate the role of each loss term on PASCAL VOC using DL3+-R101.

Table 11: Comparing with SOTA segmentation SSL methods on PASCAL VOC of various splits using DL3+-R101. All results are mIoU (%). Red: the best in a column.

| Method | 92 | 183 | 366 | 732 | 1464 |
|---|---|---|---|---|---|
| Supervise | 43.92 | 59.10 | 65.88 | 70.87 | 74.97 |
| U2PL [63] | 67.98 | 69.15 | 73.66 | 76.16 | 79.49 |
| iMAS [79] | 70.0 | 75.3 | 79.1 | 80.2 | 82.0 |
| UniMatch [71] | 75.2 | 77.2 | 78.8 | 79.9 | 81.2 |
| AugSeg [80] | 71.09 | 75.45 | 78.80 | 80.37 | 81.36 |
| JML-SSL | 76.43 | 78.47 | 79.16 | 81.21 | 82.27 |

Table 12: Ablating each loss term on PASCAL VOC using DL3+-R101. All results are mIoU (%). Red: the best in a column.

| Loss | 92 | 183 |
|---|---|---|
| $\mathcal{L}_{\text{CE, S}}$ | 43.92 | 59.10 |
| $+\mathcal{L}_{\text{CE, U}}$ | 71.09 | 75.45 |
| $+\mathcal{L}_{\text{JML, S}}$ | 72.89 | 76.20 |
| $+\mathcal{L}_{\text{JML, U}}$ | 74.74 | 77.74 |
| $+$BLS | 76.43 | 78.47 |

# J   More Results on DeepGlobe Land

We provide additional results on DeepGlobe Land in Table 13. Furthermore, we present the outcomes of BLS using DL3-R50 and KD using DL3-R18 across 5 folds of DeepGlobe Land in Table 14. Given that DeepGlobe Land is a relatively small dataset, comprising only 803 images, results for each fold can vary significantly. Nevertheless, when observing the differences between CE and JML, the mean improvement exceeds twice the standard deviation.

Table 13: Results on DeepGlobe Land (%). Best results within CE and JML groups are highlighted in red and green, respectively. Best results across CE and JML groups are underscored.

| Model | Metric | CE | CE-BLS | CE-KD | JML | JML-BLS | JML-KD |
|---|---|---|---|---|---|---|---|
| UNet-R50 | Acc ↑ | 86.89 ± 0.87 | 86.95 ± 0.99 | - | 86.59 ± 0.84 | 86.87 ± 0.79 | - |
| | mIoU ↑ | 68.80 ± 1.07 | 68.86 ± 0.81 | - | 69.29 ± 0.87 | 69.73 ± 0.89 | - |
| | ECE ↓ | 1.77 ± 0.52 | 1.68 ± 0.46 | - | 12.34 ± 1.18 | 11.68 ± 1.55 | - |
| | BECE ↓ | 21.93 ± 0.83 | 21.90 ± 1.21 | - | 40.97 ± 0.90 | 39.22 ± 1.06 | - |
| UNet-R18 | Acc ↑ | 84.99 ± 1.15 | 85.28 ± 0.97 | 85.51 ± 0.76 | 85.46 ± 0.56 | 85.53 ± 0.59 | 85.78 ± 0.66 |
| | mIoU ↑ | 64.38 ± 1.50 | 64.93 ± 0.79 | 66.40 ± 0.86 | 66.95 ± 0.25 | 66.87 ± 0.30 | 67.62 ± 0.27 |
| | ECE ↓ | 1.05 ± 0.48 | 1.47 ± 0.60 | 1.38 ± 0.61 | 10.32 ± 0.862 | 9.15 ± 0.79 | 9.83 ± 1.53 |
| | BECE ↓ | 23.23 ± 0.54 | 22.32 ± 0.75 | 25.23 ± 0.75 | 40.17 ± 0.48 | 38.13 ± 0.63 | 39.24 ± 0.94 |
| UNet-MB2 | Acc ↑ | 84.93 ± 1.11 | 84.92 ± 1.22 | 85.64 ± 1.02 | 85.54 ± 0.88 | 85.82 ± 0.66 | 85.90 ± 0.73 |
| | mIoU ↑ | 64.38 ± 1.21 | 64.03 ± 1.73 | 66.29 ± 1.39 | 66.63 ± 0.73 | 67.04 ± 0.77 | 67.46 ± 0.99 |
| | ECE ↓ | 1.64 ± 0.75 | 2.04 ± 0.74 | 1.52 ± 0.34 | 10.22 ± 1.37 | 10.01 ± 0.92 | 9.67 ± 0.86 |
| | BECE ↓ | 21.15 ± 0.89 | 20.26 ± 0.74 | 24.76 ± 1.17 | 40.40 ± 0.89 | 39.28 ± 0.27 | 38.82 ± 1.12 |

Table 14: Results of BLS using DL3-R50 and KD using DL3-R18 on 5 folds of DeepGlobe Land. All results are mIoU (%).

| Method | Loss | 0 | 1 | 2 | 3 | 4 | $\mu \pm \sigma$ |
|---|---|---|---|---|---|---|---|
| BLS | CE | 67.56 | 69.82 | 69.59 | 67.88 | 69.46 | 68.86 ± 0.81 |
| | JML | 68.13 | 70.48 | 70.60 | 69.49 | 69.93 | 69.73 ± 0.89 |
| | Diff | 0.57 | 0.66 | 1.01 | 0.61 | 1.47 | 0.86 ± 0.34 |
| KD | CE | 64.93 | 67.47 | 66.22 | 66.37 | 67.00 | 66.40 ± 0.86 |
| | JML | 67.20 | 67.96 | 67.50 | 67.62 | 67.84 | 67.62 ± 0.27 |
| | Diff | 2.27 | 0.49 | 1.28 | 1.25 | 0.84 | 1.22 ± 0.59 |

# K   More Results on Calibration

CE is a proper scoring rule [20], while the soft Dice loss is not [3]. Although the soft Dice loss can significantly increase a model's segmentation performance, it can hurt model calibration [41, 3]. We observe the same phenomenon with JML and demonstrate a potential solution by training the model with soft labels as delineated in Section 3.2. Calibration of the model can be further improved using post-hoc [20, 15, 39, 47, 52] and trainable calibration methods [31, 48]. However, these methods extend beyond the scope of this paper.

Besides, we find that models trained with JML, despite having poorer top-class calibration as quantified by ECE, actually attain superior multi-class calibration as evaluated by the static calibration error (SCE) [45]. In Table 15, we report both ECE and SCE on Cityscapes and PASCAL VOC using DL3-R101[2]. It is crucial to highlight that models trained with soft labels may exhibit worse SCE. On one side, soft labels provide an additional regularization during training, which is advantageous for calibration [44]. However, on the flip side, soft labels can represent unrealistic distributions, potentially detrimental for multi-class calibration.

Table 15: Calibration results on Cityscapes and PASCAL VOC using DL3-R101. Best results within CE and JML groups are highlighted in red and green, respectively. Best results across CE and JML groups are underscored.

| Dataset | Metric | CE | CE-BLS | JML | JML-BLS |
|---------|--------|------|--------|------|---------|
| CS | ECE (%) | 0.76 ± 0.05 | 0.97 ± 0.04 | 2.74 ± 0.03 | 2.09 ± 0.02 |
| | SCE (‰) | 2.18 ± 0.02 | 2.46 ± 0.04 | 2.08 ± 0.01 | 2.02 ± 0.01 |
| | BECE (%) | 16.14 ± 0.22 | 11.59 ± 0.12 | 30.78 ± 0.19 | 21.20 ± 0.07 |
| | BSCE (‰) | 23.81 ± 0.03 | 23.89 ± 0.04 | 23.40 ± 0.01 | 23.23 ± 0.01 |
| VOC | ECE (%) | 2.33 ± 0.03 | 2.01 ± 0.04 | 3.97 ± 0.10 | 3.20 ± 0.04 |
| | SCE (%) | 2.57 ± 0.02 | 2.59 ± 0.04 | 2.40 ± 0.06 | 2.35 ± 0.03 |
| | BECE (%) | 20.54 ± 0.12 | 17.25 ± 0.14 | 32.70 ± 0.09 | 22.30 ± 0.02 |
| | BSCE (%) | 20.94 ± 0.04 | 20.89 ± 0.01 | 20.58 ± 0.01 | 24.81 ± 0.10 |

## L  Figures

Figure 4: The best $\epsilon$ and mIoU (%) for different $k$ on PASCAL VOC using DL3-R18.

Figure 5: Effects of $\epsilon$ on PASCAL VOC using DL3-R101/50/18.

Figure 6: Code to compute active classes.

Figure 7: Qualitative results on Cityscapes.

Figure 8: Qualitative results on PASCAL VOC.

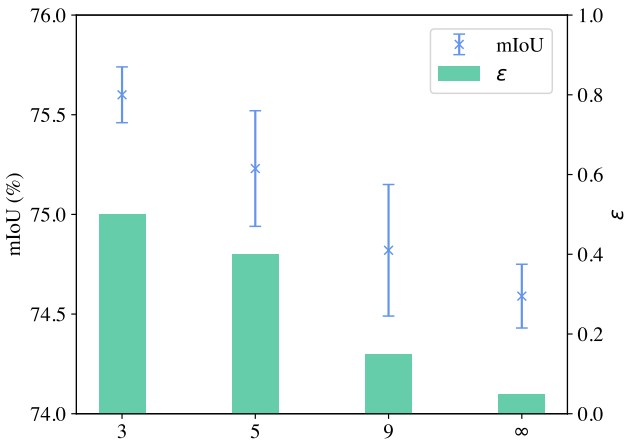

Figure 4: The best $\epsilon$ and mIoU (%) for different $k$ on PASCAL VOC using DL3-R18.

---

[2]In our code, there was an error in the computation of SCE. As a result, values reported here differ from those in our earlier arXiv version.

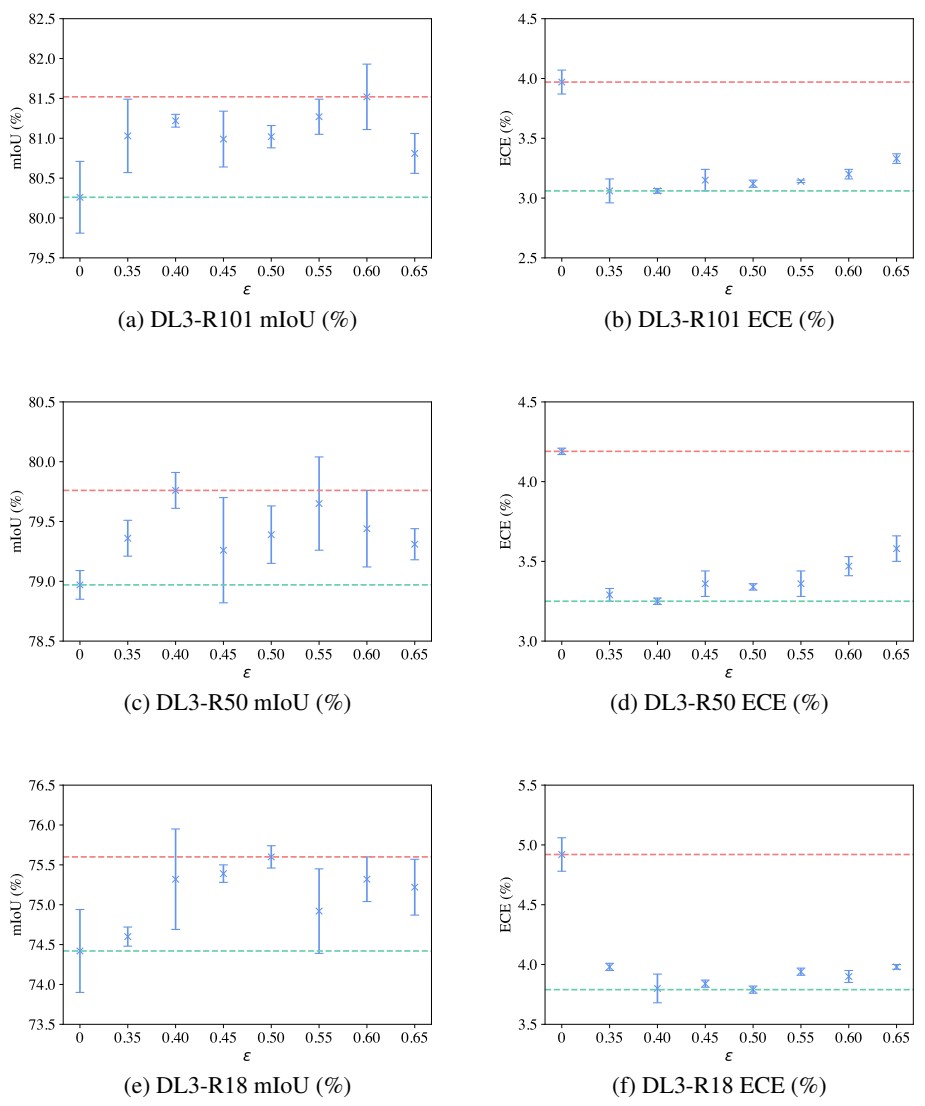

Figure 5: Effects of $\epsilon$ on PASCAL VOC using DL3-R101/50/18. $\epsilon = 0$ is the baseline (no smoothing). The highest and the lowest mean values are highlighted in red and green horizontal lines, respectively.

```
1  import torch
2  from torch.nn.modules.loss import _Loss
3
4
5  class JDTLoss(_Loss):
6      ...
7
8      # prob.shape = label.shape = (batch_size, num_classes, H * W)
9      def compute_active_classes(self,
10                                 prob,
11                                 label,
12                                 active_classes_mode,
13                                 num_classes):
14         if active_classes_mode == "ALL":
15             mask = torch.ones(num_classes, dtype=torch.bool)
16         elif active_classes_mode == "PRESENT":
17             mask = torch.argmax(label, dim=1).unique()
18         elif active_classes_mode == "PROB":
19             mask = torch.amax(prob, dim=(0, 2)) > self.threshold
20         elif active_classes_mode == "LABEL":
21             mask = torch.amax(label, dim=(0, 2)) > self.threshold
22         elif active_classes_mode == "BOTH":
23             mask = torch.amax(prob + label, dim=(0, 2)) > self.
    threshold
24
25         active_classes = torch.zeros(num_classes,
26                                      dtype=torch.bool,
27                                      device=prob.device)
28         active_classes[mask] = 1
29
30         return active_classes
```

Figure 6: Code to compute active classes. Please refer to our codebase for more details.

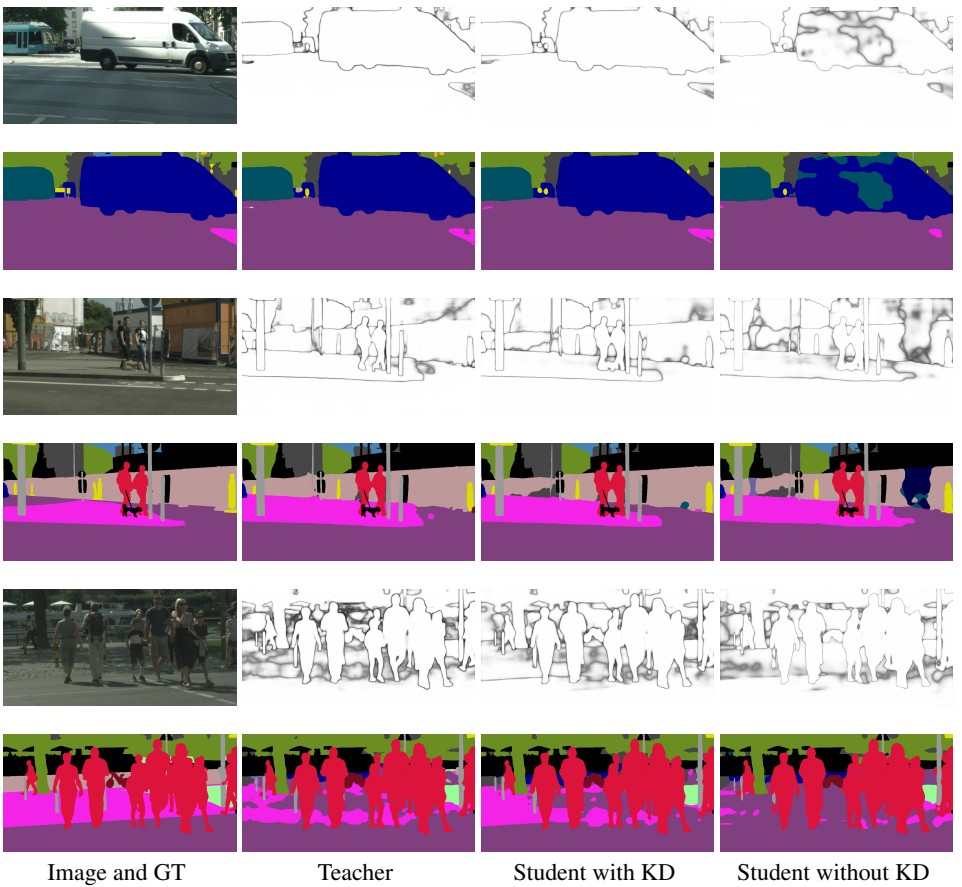

Image and GT        Teacher        Student with KD        Student without KD

Figure 7: Qualitative results on Cityscapes. Confidence maps are in black and white. Predictions are in RGB.

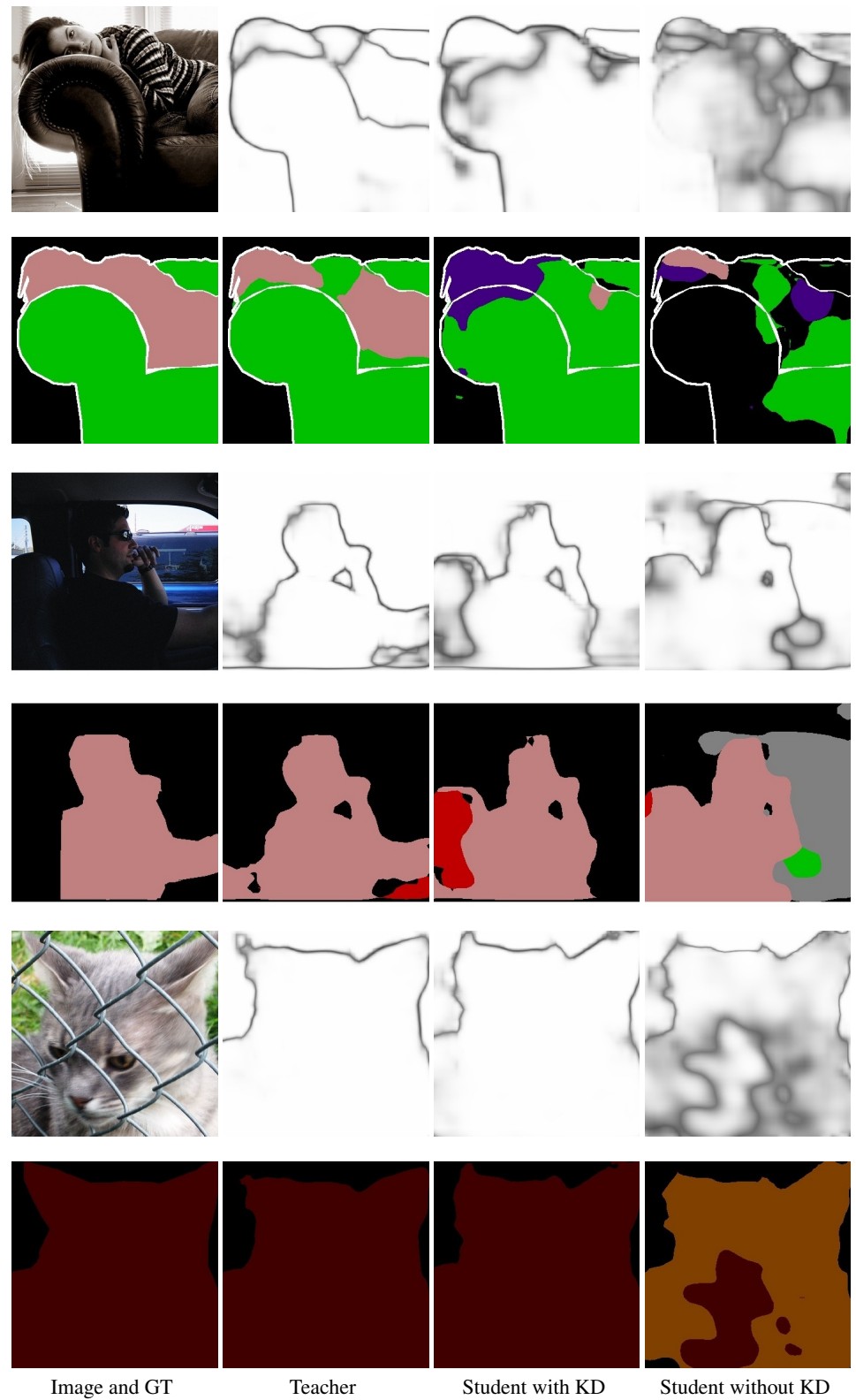

Image and GT     Teacher    Student with KD   Student without KD

Figure 8: Qualitative results on PASCAL VOC. Confidence maps are in black and white. Predictions are in RGB.

