# OpenReview forum: "Jaccard Metric Losses: Optimizing the Jaccard Index with Soft Labels"
_NeurIPS.cc/2023/Conference — NeurIPS 2023 poster_

### Official Review · Reviewer_KUTw · 2023-06-26

**Soundness:** 3 good
**Presentation:** 3 good
**Contribution:** 3 good
**Rating:** 7
**Confidence:** 5

**Summary:**

This paper presents a new loss function for semantic segmentation based on the IoU loss. The authors found that the vanilla IoU loss is incompatible with the soft labels when label smoothing and knowledge distillation training techniques are used. Experiments show that the proposed approach works well on four widely-used semantic segmentation datasets. Ablation experiments also show some insightful take-home message for the readers.

**Strengths:**

- The motivation of this paper is interesting. The authors observe an interesting problem of the vanilla IoU loss.

- The novelty of this paper is clear. Improving the vanilla IoU loss is a meaningful topic for semantic segmentation.

- The results are good. On four segmentation benchmarks, the proposed approach receives improvements.

**Weaknesses:**

- In the main paper, the authors only show results based on ConvNets. The results based on Transformers are provided in the supplementary materials. From the results, it can be seen that the improvement of the proposed approach on ConvNets are higher than that on Transformers. I think the authors should explain why this happens.

- Some visualizations should be moved to the main paper. I think it is interesting to see how the segmentation results change when the proposed method is used.

- I also would like to recommend the authors to add some failure case analysis to see what readers can further do for semantic segmentation.

- I am also interested in the results when large-sized models are used, for instance, SegFormer-B5.

**Questions:**

- Some results using Transformers should be moved to the main paper though the improvement compared to the CNNs is not that significant.

- It is also recommended to describe some works on semantic segmentation in the related work section.

- Failure cases should be provided and discussed.

**Limitations:**

Not provided.

---

> ### Author Rebuttal · Authors · 2023-08-09
>
> Thank you for your recognition of our motivation, novelty and experimental results. We address your questions about transformer experiments below.
>
> * **SegFormer-B5 experiments.** Please refer to **Table 15 (global response)**.
>
> * **Improvements on transformers are relatively lower.** We believe it is more of a limitation of BLS and KD, as reflected by relatively lower improvements in mIoU when comparing CE vs. CE-KD, and JML vs. JML-KD. Nonetheless, JML groups continue to outperform CE groups by a large margin on ADE20K using transformers (around 2\% in mIoU, even with SegFormer-B5). Moreover, we still observe a pronounced enhancement in calibration upon training with soft labels. Although our study primarily focuses on BLS, KD, and SSL, it is worth mentioning that soft labels have diverse applications beyond these, all of which could potentially gain from JML. As for the explanation, several studies [1,2,3,4] have highlighted the intricate challenges KD grapples with when integrated with transformers. For instance, to effectively utilize KD with transformers, the student model should learn from the teacher during both the pretraining and fine-tuning phases [1,2,3]. In contrast, we directly adopt pretrained weights from the timm library to avoid expensive pretraining. We will incorporate this discussion into the revised version.
>
> * **Suggestions of writing.** We value your constructive feedback on our writing and will make the necessary revisions accordingly. In particular, we will move the results on ADE20K into the main text in the revised paper.
>
> We hope that our responses can address your concerns. Please let us know if you have any follow-up comments.
>
> [1] Sanh, Victor, Lysandre Debut, Julien Chaumond, and Thomas Wolf. DistilBERT, a distilled version of BERT: smaller, faster, cheaper and lighter. NeurIPS Workshop, 2019.
>
> [2] Jiao, Xiaoqi, Yichun Yin, Lifeng Shang, Xin Jiang, Xiao Chen, Linlin Li, Fang Wang, and Qun Liu. Tinybert: Distilling bert for natural language understanding. EMNLP Findings, 2020.
>
> [3] Wu, Kan, Jinnian Zhang, Houwen Peng, Mengchen Liu, Bin Xiao, Jianlong Fu, and Lu Yuan. Tinyvit: Fast pretraining distillation for small vision transformers. ECCV, 2022.
>
> [4] Lin, Sihao, Hongwei Xie, Bing Wang, Kaicheng Yu, Xiaojun Chang, Xiaodan Liang, and Gang Wang. Knowledge distillation via the target-aware transformer. CVPR, 2022.

---

> > ### Comment · Reviewer_KUTw · 2023-08-15
> >
> > Thanks for the responses. It seems that all reviewers have recognized the contributions of this paper and the authors have solved most of my concerns. I decide to keep my original rating score unchanged.

---

### Official Review · Reviewer_83cg · 2023-06-29

**Soundness:** 4 excellent
**Presentation:** 4 excellent
**Contribution:** 4 excellent
**Rating:** 6
**Confidence:** 3

**Summary:**

The paper introduces Jaccard Metric Losses (JMLs) as a flexible alternative to Intersection over Union (IoU) losses for optimizing the Jaccard index in semantic segmentation tasks. Unlike IoU losses, JMLs can effectively process soft labels, enabling the use of important training techniques like label smoothing, knowledge distillation, and semi-supervised learning.

**Strengths:**

1. Jaccard Metric Losses (JMLs) introduce a novel approach by mirroring the soft Jaccard loss in standard settings with hard labels while remaining compatible with soft labels. This flexibility is a significant contribution as it enables the incorporation of vital training techniques that rely on soft labels, such as label smoothing, knowledge distillation, and semi-supervised learning. This novel aspect makes JMLs an interesting and valuable addition to the field.
2. The paper demonstrates good results by showcasing consistent improvements over the cross-entropy loss across multiple semantic segmentation datasets (Cityscapes, PASCAL VOC, ADE20K, DeepGlobe Land) and a variety of architectures, including both classic CNNs and recent vision transformers.
3. The paper provides theoretical analysis of JMLs, which enhances the understanding of their properties and behavior.

**Weaknesses:**

1. The paper highlights IoU losses as the baseline and the main motivation for introducing JMLs. However, the lack of direct comparisons between IoU losses and JMLs in the experiments is a weakness. While certain comparisons may not be possible, such as in knowledge distillation (KD) where IoU loss baselines require teachers, it would have been beneficial to include comparative analyses that demonstrate the importance of adapting IoU losses to soft labels. This would have provided a stronger justification for the need and effectiveness of JMLs.
2. Cross-entropy (CE) achieves better calibration performance in Tables 1 and 2. However, more discussion and explanation of this observation is needed. Understanding the reasons behind this disparity and addressing any potential limitations or trade-offs associated with JMLs' calibration performance would have added depth to the paper's analysis and interpretation.

**Questions:**

Why CE group is much better than JML group in terms of calibration?

**Limitations:**

Needs to determine hyperparameters like $\epsilon$

---

> ### Author Rebuttal · Authors · 2023-08-09
>
> Thank you for the recognition of our experiments results and theoretical analysis. We answer your questions about comparisons with other IoU losses and calibration below.
>
> * **Comparisons with other IoU losses.** Please refer to Appendix C-H. In particular, in Table 9, various losses are compared on Cityscapes and PASCAL VOC using DeepLabV3-ResNet18. We provide additional results on PASCAL VOC using DeepLabV3-ResNet101 in **Table 18 (globe response)**. In summary, JMLs are comparable with other IoU losses using hard labels, but can additionally benefit from soft labels.
>
> * **Why CE group is much better than JML group in terms of calibration?** We have a more detailed discussion of calibration in Appendix L. We appreciate your suggestion and will include additional discussions in the revised version. In particular, the reason why CE obtains better calibration than JML is because CE is a proper scoring rule [1], while SDL/SJL/JML will push predictions towards 0/1 distributions. This phenomenon is widely observed and discussed in the literature [2,3,4,5]. However, soft labels, as well as post-training techniques can greatly improve the calibration of models trained with SDL, bringing them closer to the calibration levels achieved by CE [2]. Furthermore, as highlighted in Appendix L, our findings indicate that JML attains superior calibration as measured by SCE.
>
> * **Choosing hyper-parameters (BLS $\epsilon$, JML weights).** We admit that BLS is sensitive to the choice of $\epsilon$. Nevertheless, Figure 6 (Appendix M) suggests that JML-BLS maintains solid performance across a broad range of $\epsilon$ values. Besides, please refer to our response to **Reviewer FApN** for a discussion on the selection of JML weights.
>
> We hope that our responses can address your concerns. Please let us know if you have any follow-up comments.
>
> [1] Chuan Guo, Geoff Pleiss, Yu Sun, and Kilian Q Weinberger. On calibration of modern neural networks. ICML, 2017.
>
> [2] Axel-Jan Rousseau, Thijs Becker, Jeroen Bertels, Matthew B. Blaschko, and Dirk Valkenborg. Post training uncertainty calibration of deep networks for medical image segmentation. ISBI, 2021.
>
> [3] Jeroen Bertels, David Robben, Dirk Vandermeulen, and Paul Suetens. Theoretical analysis and experimental validation of volume bias of soft Dice optimized segmentation maps in the context of inherent uncertainty. MIA, 2021.
>
> [4] Jorg Sander, Bob D. de Vos, Jelmer M. Wolterink, andIvana Isgum. Towards increased trustworthiness of deep learning segmentation methods on cardiac MRI. SPIE, 2019.
>
> [5] Alireza Mehrtash, William M. Wells, Clare M. Tempany, Purang Abolmaesumi, and Tina Kapur. Confidence Calibration and Predictive Uncertainty Estimation for Deep Medical Image Segmentation. TMI, 2020.

---

> > ### Comment · Reviewer_83cg · 2023-08-14
> >
> > Thank you for addressing my concerns.
> >
> > Forgot to mention another minor issue:
> >
> > Not sure if the use of "hard labels" is proper -- "soft labels" is easy to follow, but "hard" may be confused with "difficult".
> > Perhaps "one-hot labels" is a better term.

---

> > > ### Author Response · Authors · 2023-08-14
> > >
> > > Thanks for the suggestion. We will clearly state this distinction in the introduction.

---

### Official Review · Reviewer_FApN · 2023-07-06

**Soundness:** 3 good
**Presentation:** 3 good
**Contribution:** 3 good
**Rating:** 7
**Confidence:** 4

**Summary:**

The paper proposes JML (Jaccard Metric Losses) for segmentation. This builds up on previous IoU-based losses but with the added ability to process soft labels or probability distributions. JML losses can be further applied as follows:
JML-BLS (Label smoothening Jaccard-based IoU loss, which is only computed at boundary pixels based on empirical analysis).
JML-KD (Applying the soft JML loss to between student and teacher, as well as applying the soft JML loss between student and GT).

**Strengths:**

Presentation:
The reviewer appreciates authors' clarity of presentation of their concept. The paper is quite clear and tables are easy to understand. Mathematical notations have been thoroughly reviewed and hence the paper reads well.

Originality:
Although the work takes inspiration from IoU loss, and is quite a simple update to the original IoU loss concept, the technical contribution is quite clear and holds up. There has been work on IoU losses before but they are not used widely in literature, hence, there is a scope for improvement in this area. This work is a beneficial building block to further research in IoU losses.

Transparency:
The reviewer appreciates the code being made available.

Results:
The JML loss seems to work on a wide range of datasets, and more importantly, backbones. There are 12 different architectures in the paper as well as supp, and the loss works for both CNNs as well as transformer based backbones. This is very interesting as it shows the capability of this loss to provide a high impact in our field.



**Weaknesses:**

There is a lot of information in the appendices, which might be more beneficial than some of the things in the main paper. One such example is the experiment with transformer-based backbones.

**Questions:**

1. In tables 1-13, Did each experiment (row) start with the same set of pre-trained weights?
2. How did you tune the hyper-parameters for JML losses (specifically loss weights)? How easy(or difficult) is it to obtain the results provided in the paper.
3. Does adding JML ever reduce performance (with a higher weight, etc)
4. Could you address limitations of this work

**Limitations:**

No - The authors are suggested to address the limitations of this method (if any).

---

> ### Author Rebuttal · Authors · 2023-08-09
>
> Thank you for your appreciation of the originality, transparency, and potentially high impact of our work. We are encouraged. We answer your questions about experimental setups and hyperparameter tuning as follows.
>
> * **Experimental setups.** We assure that each experiment starts with the same set of pre-trained weights, with the exception of those in Table 3, 4 where we adopt numbers reported in those SOTA papers. Since different SOTA methods usually adopt diverse codebases, pretrained weights and training hyper-parameters, these tables cannot offer a completely fair comparison. However, as noted in line 194-200, our checkpoints are from the standard timm library pretrained on vanilla ImageNet (not ImageNet21K). Furthermore, as outlined in line 224-229, our baseline student model does not unfairly (e.g. longer training epochs, larger crop size, pre-training on larger datasets, etc) outperform those in other KD methods.
>
> * **How did you tune the hyper-parameters for JML losses?** We use a 0.25/0.75 ratio to mix CE and JML, following the suggestion in MMSegmentation. The ratio of 0.5/0.5 is also widely used in many segmentation papers, e.g. DETR, Mask2Former, SAM, etc.
>
> * **Does adding JML ever reduce performance?** According to your question, in **Table 17 (global response)**, we add ablation studies for JML weights on PASCAL VOC using both DeepLabV3-ResNet101 and DeepLabV3-ResNet18. Compared with using CE alone (the rightmost column), adding JML will not reduce performance across various loss weights.
>
> * **Could you address limitations of this work?** Thank you for drawing attention to this important aspect. We recognize several limitations and future research directions, and we will add this discussion into our revised manuscript. To elucidate:
>   * Currently, we only demonstrate JMLs' effectiveness as a metric in the label space. Nevertheless, in many applications or techniques, feature-space metrics are beneficial, e.g., the feature distillation technique. Whether JML can be successfully extended to the feature space is worth studying.
>   * Our application of JMLs is predominantly in semantic segmentation. However, given the exceptional efficacy of JMLs in optimizing Acc (as detailed in lines 210-214), we anticipate their applicability even in standard classification tasks. This is especially relevant in scenarios with data imbalance, as recently delved into by the NLP community [4].
>   * While our current study concentrates on three soft label applications (LS, KD, SSL), there are a plenty of scenarios incorporating soft labels that worth deeper exploration. This includes areas such as neural architecture search [1], dynamic inference [2], and adversarial machine learning [3], among others.
>
> * **Suggestions for writing.** Thank you for your suggestion. We will add the results on ADE20K into the main text of the revised paper.
>
> We hope that our responses can address your concerns. Please let us know if you have any follow-up comments.
>
> [1] Li, Changlin, Jiefeng Peng, Liuchun Yuan, Guangrun Wang, Xiaodan Liang, Liang Lin, and Xiaojun Chang. Block-wisely supervised neural architecture search with knowledge distillation. CVPR, 2020.
>
> [2] Yu, Jiahui, and Thomas S. Huang. Universally slimmable networks and improved training techniques. ICCV, 2019.
>
> [3] Papernot, Nicolas, Patrick McDaniel, Xi Wu, Somesh Jha, and Ananthram Swami. Distillation as a defense to adversarial perturbations against deep neural networks. S&P, 2016.
>
> [4] Li, Xiaoya, Xiaofei Sun, Yuxian Meng, Junjun Liang, Fei Wu, and Jiwei Li. Dice loss for data-imbalanced NLP tasks. ACL, 2020.

---

> > ### Comment · Reviewer_FApN · 2023-08-17
> >
> > Thanks for addressing my questions. I keep my original rating.

---

### Official Review · Reviewer_L5GW · 2023-07-06

**Soundness:** 3 good
**Presentation:** 3 good
**Contribution:** 3 good
**Rating:** 5
**Confidence:** 3

**Summary:**

The paper focuses on making IoU losses flexible to process soft labels. Jaccard Metric Losses are presented that works the same as soft Jaccard loss with hard labels but it is also compatible with soft labels. Experiments are presented in several tasks: label smoothing, knowledge distillation, and semi-supervised learning. The results show improvement over baselines.

**Strengths:**

+ The paper focuses on addressing an important issue with IoU losses that they are not flexible to process soft labels
+ Experiments on several tasks (i.e., label smoothing, knowledge distillation, and semi-supervised learning) shows promising results.

**Weaknesses:**

There are several confusions regarding experimental results.

--There are discrepancies between the results presented in Table 5 and Table 3 for Cityscapes with DL3-R18. Table 3 reports 77.91 ± 0.16 whereas Table 5 reports 76.68 ± 0.33 as the best performance. For VOC, the best results match in the tables (75.89 ± 0.29). However, there is confusion between JML-KD and JML-BLS. For VOC, the JML-KD results in Table 3 match JML-BLS results in Table 5.

-- As shown in Table 3, in knowledge distillation proposed student achieves 78.14. This is a bit surprising as the Teacher DeepLabV3-R101 has a lower mIoU of 78.07 (as reported in DIST[21]).

-- The reported performance in Table 1 and Table 2 do not compare with SOTA performance as shown in the papers. For example, the deeplabv3  paper with DL3-R101 model reports 81.3 mIoU in CityScapes and 85.7 mIoU in Pascal VOC. The presented proposed method results are lower compared to that and baselines are lower too. We see this for all the cases in Tables 1 and 2. I would suggest the authors use the proposed loss with the best-performing model and losses from the prior works to avoid any confusion.  To better convince the readers,  I think improving on the best results reported in prior works would be helpful.

**Questions:**

Please address the concerns in the weaknesses section.

-- The rebuttal addressed most of my concerns. Hence, I increase the score.

**Limitations:**

yes

---

> ### Author Rebuttal · Authors · 2023-08-09
>
> Thank you for the recognition of our motivation and promising results over several tasks. We answer your questions with respect to experimental settings as follows. Please also refer to our response to **reviewer FApN** for a more discussion of experimental setups.
>
> * **There are discrepancies between the results presented in Table 5 and Table 3 for Cityscapes with DL3-R18.** In all tables except for Table 3, we follow the training setting as mentioned in Appendix B. In particular, we use a batch size of 8 in Table 1 and 5. Training on Cityscapes is very expensive due to the large crop size. However, in order to match the training setting in SOTA segmentation KD papers (CIRKD, MasKD, DIST), as mentioned in the caption of Table 3, we increase the batch size from 8 to 16 solely for Cityscapes experiments. Therefore, the performance of DeepLabV3-ResNet18 on Cityscapes in Table 3 is higher than those in Table 1 and 5. We will ensure this distinction is highlighted in the revised version of the paper.
>
> * **Table 5.** This table is designated for an ablation study of JML-KD. In Table 5, we isolate the contribution of each proposed term, which are defined in section 2.4 and explained in the caption. In particular, $\mathcal{L}_{\text{JML-BLS}}$ means we train the teacher with JML-BLS. Hence, in Table 2, 3, 5, the results of JML-KD on PASCAL VOC using DeepLabV3-ResNet18 should match, i.e. $75.89 \pm 0.29$.
>
> * **Teacher of DIST vs. ours.** Their teacher is DeepLabV3-ResNet101 trained with CE (78.07), while ours is DeepLabV3-ResNet50 trained with JML-BLS ($79.10\pm0.35$). We choose ResNet50 instead of ResNet101 as the teacher to save the computational cost, and the reason for training the teacher with JML-BLS is described in line 187-192 and ablated in Table 5, 7.
>
> * **Results of DeepLabV3-ResNet101 in [1] vs. ours.** In [1], 81.3 on Cityscapes is on the test set, with flipping inference and multi-scale inference. Their results on the val set, without flipping inference and multi-scale inference, is 77.82. 85.7 on PASCAL VOC is also on the test set, with flipping inference, multi-scale inference, and COCO pretraining. Their results on the val set, without flipping inference, multi-scale inference, and COCO pretraining, is 78.05. We do not include these extra ingredients, i.e. inference strategies (flipping, multi-scale) and COCO pretraining, and report the results on val set, following and for a fair comparison with other segmentation papers considered in our study (SKD, IFVD, CWD, CIRKD, MasKD, DIST, U2PL, iMAS, UniMatch, AugSeg). For example, the performance of DeepLabV3-ResNet101 on Cityscapes reported in DIST is 78.07. Ours trained with CE on Cityscapes val set and PASCAL VOC val set are $78.67 \pm 0.32$ and $78.39 \pm 0.09$, respectively.
>
> * **Comparing with SOTA results.** In **Table 16 (global response)**, we compare with MMSegmentation, which provides a large-scale benchmark of various segmentation models, on Cityscapes, PASCAL VOC and ADE20K. We obtain comparable results on Cityscapes and PASCAL VOC using DeepLabV3-ResNet101. Note that on ADE20K, SegFormer-B3 is trained with 160K iterations in MMSegmentation, while ours is only 40K.
>
> We hope that our responses can address your concerns. Please let us know if you have any follow-up comments.
>
> [1] Liang-Chieh Chen, George Papandreou, Florian Schroff, and Hartwig Adam. Rethinking Atrous convolution for semantic image segmentation. arXiv, 2017.

---

> ### Author Response · Authors · 2023-08-20
> **Thanks for the review! We'd like to make sure if our responses have addressed your concerns**
>
> Dear reviewer,
>
> We really appreciate your time in reviewing our work. As the discussion period draws to a close, we'd like to take the chance to kindly inquire whether our responses have addressed your concerns. We're available to answer further concerns if there is still something unclear.
>
> Best,
> From the authors

---

### Official Review · Reviewer_jE79 · 2023-07-07

**Soundness:** 3 good
**Presentation:** 3 good
**Contribution:** 2 fair
**Rating:** 5
**Confidence:** 3

**Summary:**

In this paper, the authors propose Jaccard Metric Losses (JMLs) for the training with soft labels in semantic segmentation. The authors analyze the limitations of previous IoU losses for soft labels, and propose JMLs to solve such limitations. Experiments for label smoothing, knowledge distillation and semi-supervised learning demonstrate the effectiveness of JMLs.

**Strengths:**

1. It is interesting that using Jaccard-based losses to solve the soft label training problem.
2. This paper is well written and easy to read.
3. Experiments show large improvements for multiple architectures.

**Weaknesses:**

1. The models in experiments seem a bit old. It would be great to show the performance with more recent architectures, such as Mask2Former.
2. Only small models are compared in experiments. It is important to show the performance for larger models to demonstrate the scalability.


**Questions:**

Please solve my concerns in Weaknesses.

---

> ### Author Rebuttal · Authors · 2023-08-02
>
> Thank you for the evaluation that our paper is interesting, well-written, and has large improvements over multiple architectures. We respond to your concerns in terms of architecture selections as follows.
>
> * **The models in experiments seem a bit old.** In the main paper, we mainly experiment with DeepLabV3, DeepLabV3+ and PSPNet. We select these models to be in line with SOTA segmentation KD papers (SKD, IFVD, CWD, CIRKD, MasKD, DIST) and SOTA segmentation SSL papers (U2PL, iMAS, UniMatch, AugSeg). Moreover, the reviewer could kindly refer to Appendix J, where we present results on ADE20K using SegFormer (NeurIPS2021, ~1500 citations), which is one of the most popular and recent segmentation models.
>
> * **Mask2Former (CVPR2022, ~500 citations).** Thanks for this specific suggestion. We'd like to share our considerations here. This model represents a significant deviation from other mainstream segmentation models such as DeepLabV3, DeepLabV3+, PSPNet, SegFormer, UPerNet, UNet, etc. Its innovation does not only lie in its architecture design (masked attention), but also in its training settings (the Hungarian algorithm and mask loss). These distinctive features aim to unite semantic/panoptic/instance segmentation, but unavoidably make training on each individual task more complex. To evaluate JMLs' potential in a more clean setting and avoid additional compounding factors, our evaluation does not include this specific method design. We'll add the discussion to our paper.
>
> * **Only small models are compared in experiments.** Thanks for this comment and suggestion. We have experimented with DeepLabV3-ResNet101 and DeepLabV3+-ResNet101. These models are fairly large. Also, just for some reference, numerous excellent segmentation literature, including the baselines in our paper (SKD, IFVD, CWD, CIRKD, MasKD, DIST, U2PL, iMAS, UniMatch, AugSeg) have not experimented with larger models. To position the ability of JML in this vaster literature, we select these models for comparison. Moreover, to echo your valuable concern and further underscore the effectiveness of JML on larger models, in **Table 15 (global response)**, we deliver additional results on ADE20K using SegFormer-B5, which is the largest model in the SegFormer family.
>
> We hope our responses can solve your concerns. Please let us know if you have any follow-up comments.

---

> > ### Comment · Reviewer_jE79 · 2023-08-18
> >
> > Dear authors,
> >
> > Thanks for your reply.
> >
> > Despite high citations, DeepLabV3+, PSPNet and SegFormer are actually old baselines in the segmentation community.
> >
> > Although Mask2Former may be more complex than previous methods, it and its modifications (such as ViT-Adapter and Oneformer) are the most popular semantic segmentation paradigms after 2022. They are quite different from previous baselines and achieve much higher performance. Therefore, I think it is important to show the effectiveness on this paradigm.
> >
> > In my opinion, if a loss can only be used in architectures 2 years ago, it might be hard to help future research.

---

> > > ### Author Response · Authors · 2023-08-18
> > > **Thanks for the reviewer's further comments**
> > >
> > > Many thanks for your follow-up comments! Now, we better understand the suggestion: In addition to the demonstrated practical usage on DeepLabV3+, SegFormer, etc, it would be great to demonstrate JMLs' effectiveness on Mask2Former to raise more future research interests. We're starting the experiments to demonstrate whether JMLs' ability to leverage soft labels can help the newest architectures.
> > >
> > > Before providing the results, currently, we can only provide the simple reasoning:  As we all know, Mask2Former uses Soft Dice Loss, which is similar in spirit to Soft Jaccard Loss (SJL) and yield similar results [1]. And our experiments have demonstrated that JML can consistently improve SJL across a wide range of architectures. Therefore, we expect JML can help leverage the information in soft labels to further improve Mask2Former. After we got the results, we'll add the results to the revision of the paper.
> > >
> > > [1] Tom Eelbode, Jeroen Bertels, Maxim Berman, Dirk Vandermeulen, Frederik Maes, Raf Bisschops, and Matthew B. Blaschko. Optimization for medical image segmentation: Theory and Practice When Evaluating With Dice Score or Jaccard Index. TMI, 2020.

---

> > > > ### Comment · Reviewer_jE79 · 2023-08-21
> > > >
> > > > Thank you. I will keep my original rating. Overall, this work is well organized. I really hope the experiments can be improved in the revision.

---

### Author Rebuttal · Authors · 2023-08-09

We thank all reviewers for your time and efforts. The questions and suggestions are very valuable for us to improve our work. In this global response, we attach a one-page pdf that contains additional experiments.

---

### Decision · Program_Chairs · 2023-09-21

**Decision:**

Accept (poster)

**Comment:**

The paper receives four positive ratings and one borderline rejection. The AC took a close look at the paper, reviews, and rebuttals. Specifically, for reviewer L5GW who provided the negative review, the AC finds that the raised questions are addressed well.

The paper demonstrates an effective loss function for semantic segmentation, which is applicable for various techniques, e.g., label smoothing, distillation, and semi-supervised learning. The authors also show extensive experiments with various model architectures and benchmark datasets. Therefore, the AC recommends to accept this paper.